# Generating particle physics Lagrangians with transformers

**Yong Sheng Koay**[1*]**, Rikard Enberg** [1]**, Stefano Moretti** [1,2] **and
Eliel Camargo-Molina** [1,3†]

**1** Department of Physics and Astronomy, Uppsala University, Sweden
**2** School of Physics and Astronomy, University of Southampton, UK
**3** Department of Game Design, Uppsala University, Sweden

★ yongsheng.koay@physics.uu.se ,    † eliel.camargo-molina@physics.uu.se

## Abstract

In physics, Lagrangians provide a systematic way to describe laws governing
physical systems. In the context of particle physics, they encode the inter-
actions and behavior of the fundamental building blocks of our universe. By
treating Lagrangians as complex, rule-based constructs similar to linguistic
expressions, we trained a transformer model — proven to be effective in nat-
ural language tasks — to predict the Lagrangian corresponding to a given
list of particles. We report on the transformer's performance in construct-
ing Lagrangians respecting the Standard Model $SU(3) \times SU(2) \times U(1)$ gauge
symmetries. The resulting model is shown to achieve high accuracies (over
90%) with Lagrangians up to six matter fields, with the capacity to generalize
beyond the training distribution, albeit within architectural constraints. We
show through an analysis of input embeddings that the model has internal-
ized concepts such as group representations and conjugation operations as it
learned to generate Lagrangians. We make the model and training datasets
available to the community. An interactive demonstration can be found at:
https://huggingface.co/spaces/JoseEliel/generate-lagrangians.

## Contents

# 1   Introduction

A Lagrangian is an object that describes the dynamics of physical systems in terms of their kinetic and potential energies, with equations of motion obtained by applying the Euler–Lagrange equations to it. While born out of classical mechanics formulations, it is of central importance in Quantum Field Theory (QFT), which is the language of all of modern theoretical physics, whichever the discipline (whether particle physics or others). The Lagrangian encodes the detailed dynamics of a physics model, as well as the symmetries obeyed by the system under study. This means that a central task in physics is to understand how a Lagrangian is constructed and make sure that it reflects the symmetries of the system.

Transformers originated [1] with translation tasks and Natural Language Processing (NLP) and are now essential for Large Language Models (LLMs) but also generative models more generally. Their main feature is the mechanism of self-attention. When translating from one natural language to another, it is necessary to pay attention to other words in the sentence since words are context-dependent or, in other terms, there are long-range correlations in a sentence. This is achieved through self-attention, which essentially means that the transformer 'transforms' a sequence of input vectors of tokens into an output sequence that mixes these tokens with suitable weights. By doing so the context of a whole

sentence can be encoded in a way that contains information about the relation between words.

The sequence of tokens in LLMs represents words and punctuation in a sentence or set of sentences while, in this paper, we will apply a generative transformer architecture to QFT Lagrangians and the tokens will instead represent the parts of their expressions: the different factors that make up the terms of the latter (the fields) and the mathematical operators that combine these together (their interactions). One can draw parallels between linguistic structure and that of Lagrangians, where fields, their combinations (terms) and the invariance of the whole expression under symmetry resemble words, sentences and grammatical structures, respectively. Similar to how a well-constructed paragraph follows grammatical rules to convey meaning, a Lagrangian obeys symmetries that govern relationships between different fields and terms that express the dynamics of particle interactions. And just like how a single word of a sentence can change the overall meaning of the sentence, a single field can change the interactions between particles and a single term in the Lagrangian can change the properties that it gives to the model, starting from its cosmological evolution to predictions for present day experiments at, e.g., the Large Hadron Collider (LHC).

Generating the Lagrangian for a given set of fields can be accomplished by humans or computers and, most often in practice, a combination of the two. While achieving perfect accuracy using a transformer model would be remarkable, it is not our primary goal here. Instead, we focus on how transformer models can learn aspects of symbolic mathematics, which is valuable per se, i.e., well beyond just generating Lagrangians. The code we use to generate our dataset can write Lagrangians from a list of particles automatically, which at first might seem like it defeats the purpose of a training a transformer model with data generated by it. But this is the first step of a long-term objective to build a foundation model capable of identifying, processing, manipulating and exploring equations, indeed, a basis for theoretical physics. This model could find broader applications just as NLP has evolved from text completion to translation, assistants and beyond. A medium-term goal is a transformer-powered system to explore theories Beyond the SM (BSM) of particle physics, the latter being the prevalent description of Nature at its most fundamental level.

Transformers have already found significant use in High Energy Physics (HEP) for data-driven tasks such as jet tagging, anomaly detection, data analysis and lattice calculations, see, e.g., [2–25], but there has been a lack of application in more theoretical contexts like symbolic mathematics. This work aims to fill that gap by applying transformers to an area that transcends traditional data analysis, which is intrinsically *numerically* based, to instead address a form of structure learning, which is intrinsically *symbolically* based. In symbolic manipulation, Machine Learning (ML) methods have shown promise. Works on symbolic regression [26, 27] and Deep Learning (DL) applications to mathematical problems [28] provide a basis for our approach. Work has been done on applying transformers on scattering amplitudes [29–32] and linear algebra [28, 33, 34]. Other approaches for building models, based on Generative Adversarial Networks (GANs) or Reinforcement Learning (RL) have also been explored [35–41], but none exploiting transformer architectures. In this work, the focus is on symbols (such as fields, covariant derivatives, Dirac symbols, etc.) that implicitly or explicitly encode information about all the *symmetries* involved in their interactions, in turn determining which terms in the Lagrangian are allowed and which are not. Therefore, this is a step forward in the application of transformer models in symbolic scientific work. In fact, we go beyond previous works tackling less complex mathematical expressions such as *arithmetic*, *linear algebra*, or *calculus*, where symbols are either simple *numerical values* (e.g. $1$, $-7$, $1/2$, $3.4$, $-3\mathrm{e}6$,...), *symbolic variables* (e.g. $x$,$y$,...), or *mathematical operators* $(+, -, \times, /, {}^2, \sqrt{}, \log, d/dx, \int, \dots)$ where symmetries (if any) are only manifested when the

expression/functions are defined. In other words, a single gauged fermion field symbol $\psi_L$ contains more information than a single variable $x$ and a covariant derivative symbol $D_\mu$ contains more information than a single differential operator $d/dx$. One of the more recent examples, which tackles scattering amplitudes [29], shows the growing interest in going beyond simpler symbolic expressions by incorporating certain symmetry elements (e.g., Lorentz symmetry through spinors and gamma matrices). While this work is an important milestone and evidence that approaches like ours are a logical next step, it still does not encompass the full scope of symmetry-rich fields and operators we address here.

The paper is structured as follows. In the next section, we explain the role of Lagrangians in particle physics. This is followed by a section where we describe the construction of our dataset. In Section 4, we describe the training stage of our model and its performance in in-distribution dataset. In Section 5, we explore the model's capabilities to generalize in Out-Of-Distribution (OOD) data. In Section 6 we assess what our model has learned. We then share some information on how to access and operate our model and datasets in 7.

## 2   Lagrangians in particle physics

The Lagrangian formalism is the cornerstone of modern theoretical physics. It is a mathematical framework that describes the dynamics of physical systems in terms of their kinetic and potential energies, encapsulated in a function called the Lagrangian and denoted by $\mathcal{L}$.

The equations governing the dynamics of a system can be derived from its Lagrangian using the principle of least action, which states that the path taken by a system between two points in spacetime is the one that minimizes (or extremizes) the action $S$, or the integral of the Lagrangian over spacetime[1]:

$$S = \int \mathcal{L} \, d^3x \, dt \tag{1}$$

where $\mathcal{L}$ is the Lagrangian, $t$ is time and $d^3x$ is the volume element in three-dimensional space.

In the realm of particle physics, Lagrangians play a critical role in formulating QFTs, where they provide a systematic way to describe the interactions and behavior of fundamental particles.

Constructing valid Lagrangians is a complex task. They are built from differential operators and quantum fields that respect a set of symmetries. The fields may be scalars, spinors, vectors, or in general tensors of different ranks, which are functions of spacetime coordinates. The rules governing their construction are intricate, involving considerations of symmetry invariance requiring the correct combination of field components. The general principle is that the Lagrangian contains *all* possible terms involving the considered fields that are invariant under the considered symmetries, and *only* those terms.

In this section we provide a brief overview of some theoretical aspects of particle physics, focusing on the construction of Lagrangians. If the reader is already familiar with this topic, they may skip to the next section.

---

[1] In classical mechanics, the Lagrangian $L$ is a function of the system's coordinates and velocities and depends only on time. The action $S$ is then obtained by integrating $L$ over time: $S = \int L \, dt$. In contrast, in QFT, the Lagrangian $\mathcal{L}$ depends on fields that are functions of both space and time (or of spacetime). Consequently, the action is obtained by integrating the Lagrangian *density* $\mathcal{L}$ over spacetime. The quantity $\mathcal{L}$ is properly called the Lagrangian density, but in this article we will refer to it simply as the Lagrangian, following the usual convention in particle physics.

## 2.1   Symmetries

Symmetries are fundamental in physics, providing deep insights into conservation laws and dictating the interactions between particles. In physics, a *symmetry* refers to a transformation that can be applied to a system without changing its observable physical properties. These transformations are mathematically described in *group theory* by structures called groups, or symmetry groups. The different fields in the Lagrangian are transformed by elements of these symmetry groups, in various representations of the groups: we say that the fields transform under a certain representation of the group, which essentially means that the field is an element of an abstract vector space on which the elements of the group act as linear operators.

    The symmetry groups are crucial for ensuring that the physical laws derived from the Lagrangian are consistent with the symmetries of the system. The logic is that when a certain symmetry of nature is imposed, then all fields transform under some irreducible representation of the corresponding symmetry group. If the field is not affected by these transformations, we say that it transforms under the trivial representation, or that it transforms as a singlet, which we here denote as **1**. Otherwise, it transforms under a larger, non-trivial representation. The point is that if nature is invariant under a certain symmetry, then the Lagrangian must also be invariant under that symmetry, meaning that it must only contain combinations of fields that are invariant under symmetry transformations.

    Mathematically, each term in the Lagrangian is a tensor product of elements of the representations of each field entering the term. The complete term built out of fields must then transform as a singlet under the group, which mathematically means that, first, the reducible representation built out of the tensor product of fields must contain a singlet representation and, second, the explicit way that the fields are combined must transform under that singlet representation. Concretely, this is done by writing each term in tensor notation, with implicit summations over indices according to the Einstein summation convention. If all indices are summed over, or *contracted*, in the correct way, then the resulting term is invariant under the symmetry group—this is similar to how dot products of vectors produce scalars that are invariant under rotations. We will make heavy use of such contractions below.

    A QFT might have symmetries built out of several groups, each group representing a different aspect of the theory. The SM of particle physics is our best description of the fundamental particles and forces in the universe so far. In the SM, gauge symmetries (the symmetries associated with fundamental forces) are described by the *gauge group* combination $SU(3)_C \times SU(2)_L \times U(1)_Y$, which is a direct product of three unitary groups:

- $SU(3)_C$: the gauge group of the strong nuclear force, associated with color charge;

- $SU(2)_L$: the gauge group of the weak nuclear force, associated with weak charge;

- $U(1)_Y$: the abelian gauge group associated with hypercharge.

Note that in the SM, the gauge group is spontaneously broken by the vacuum expectation value of the Higgs field down to $SU(3)_C \times U(1)_{em}$, where the last $U(1)$ factor is not the same as the $U(1)_Y$, but rather the electromagnetic gauge group. We do not consider spontaneous symmetry breaking in this paper.

    Many extensions of the SM, such as Grand Unified Theories (GUTs) and Supersymmetric (SUSY) models, are built on the same principles, but with different symmetries. Physicists are interested in extending the SM to describe phenomena like dark matter, dark energy and the unification of forces, which are not explained by the SM. In this paper we will restrict ourselves to the gauge group of the SM, although we will consider fields that are in other representations of the groups than the ones that appear in the SM.

185     In addition, particle theories are built to be consistent with special relativity, which
186 means the system has to respect an extra symmetry group, the *Poincaré group*, which is an
187 extension of the Lorentz group SO(1,3) that also describes symmetries under translations.
188 These are spacetime symmetries.

189     Other symmetries can also be present, such as global symmetries that are not associated
190 with forces, but with conservation laws, or discrete symmetries such as CP symmetry or the
191 *R*-parity of supersymmetric theories. In this paper we will not consider any such additional
192 symmetries.

## 193   2.2   Fields and invariants

194 In QFTs, particles are represented by quantum fields. Each fundamental particle corre-
195 sponds to a specific field. For example, the electron is described by the electron field and
196 quarks by quark fields. Fields are organized based on their transformation properties, i.e.,
197 representations, under the various symmetries that leave the Lagrangian invariant.

198     To encode which representations a field belongs to in our mathematical expressions,
199 each field is assigned labels that determine how they transform under each component
200 of the gauge group and other symmetries. More specifically, each field that we consider
201 transforms under a specific irreducible representation of each of the three gauge groups
202 that make up the SM gauge group. As is common practice, we label the representations of
203 these groups by their dimension (in boldface) for the $SU(3)_C$ and $SU(2)_L$ groups, and by
204 their hypercharge $Y$ (a rational number) for the $U(1)_Y$ group, since $U(1)$ is just the group
205 of phase factors, $\phi \rightarrow e^{iY\theta}\phi$.

206     As explained above, each term in the Lagrangian must be arranged in such a way that
207 it transforms as a singlet under the total SM gauge group, which means that it must be a
208 singlet under each of the three component groups. That puts strong restrictions on which,
209 and how many, fields of each type that can be combined.

210     The different terms in the Lagrangian can be classified into three types of terms: *Kinetic*
211 *terms* are those with two fields and two derivatives, and *mass terms* are those with two
212 fields and a mass parameter. These two terms together are used to describe how a particle
213 propagates through spacetime. Finally, *interaction terms* are those with three or more fields
214 and are used to derive the Feynman rules that describe interactions.[2] This classification of
215 terms is not unambiguous, but will be useful. In particular, we will study how trilinear and
216 quartic terms are generated by the transformer model. These correspond to how interaction
217 vertices with three or four particles are described by the theory.

218     In this paper, we will consider the following representations[3].

219     • $SU(3)_C$: We will consider singlets, denoted by **1**, and fields in the fundamental or
220        anti-fundamental three-dimensional representations **3** and **3̄**. In the SM, quarks are
221        in the **3** and anti-quarks in the **3̄**. All other fields are singlets under $SU(3)_C$.

222     • $SU(2)_L$: We will consider singlets **1**, doublets **2**, and triplets **3**. Note that $SU(2)$ is
223        special in that a representation and its complex conjugate representation are equivalent
224        so we do not distinguish these. In the SM, left-handed leptons and quarks as well as
225        the Higgs field are doublets, while right-handed leptons and quarks are singlets.

226     • $U(1)_Y$: we will consider a range of rational numbers, as explained below. In the SM,
227        a wide array of hypercharges appear.

---

[2]Examples of interaction terms and mass terms can be found in Appendix C

[3]Note that we do not explicitly include gauge fields, which are always in the adjoint representation of
the gauge group, which is $(N^2 - 1)$-dimensional for $SU(N)$ and $N^2$-dimensional for $U(N)$.

The fields can be combined in several different ways to obtain singlets under the gauge group. In Appendix A we show explicit examples, but the main result is that for $SU(3)_C$, the product of a triplet and an antitriplet can yield a singlet, while two triplets can not. Three triplets can also form a singlet. For $SU(2)_L$, singlets can be formed from products of two doublets, two triplets, two doublets and one triplet, or three triplets. There are also combinations of four fields that can form triplets.

The Lagrangian furthermore must respect special relativity. Each field transforms under an irreducible representation of the Poincaré group, and each term must again be a singlet. Irreducible representations of the Poincaré group are labeled by their mass and spin, and the relevant quantity for obtaining singlets is the spin. In this paper we consider spin-0 fields and spin-1/2 fields, which are singlets and doublets, respectively, under the Poincaré group, which means that they are scalar fields or fermion fields. Scalar fields are invariant by themselves, while fermion fields must be combined with other doublets to form a singlet combination. Thus each term must contain an even number of spin-1/2 fields, but they can be combined in several different ways, i.e., their indices can be contracted in different ways. For example, the indices on fermions associated with the Lorentz group are contracted using special matrices called the Dirac $\sigma$ matrices. An explicit example of this is shown later in this section. See also e.g. [42] for a pedagogical discussion. However, there is no restriction on the number of scalar fields from special relativity. It is therefore "harder" to obtain an invariant trilinear interaction term with fermions than with scalars.

Finally, we impose the constraint that each term must be renormalizable. We will not discuss this requirement here, but it can be formulated as a constraint on the mass dimension of each term, see e.g. [42, 43] for a discussion. For our purposes this leads to a limitation on the number of fields in each term: We may only have terms with two to four scalar fields, two fermions, or two fermions and one scalar.

In the following, we collectively refer to the labels specifying the spin, helicity, color, weak isospin, hypercharge, etc., which together define how each field transforms under the symmetries, as its "quantum numbers".

## 2.3 Building invariant Lagrangians

In order to build the most general Lagrangian, as explained above, we must include all terms that are invariant under the imposed symmetries. We must therefore systematically explore all the ways in which the available fields can be multiplied and combined, for all the symmetries mentioned here.

To describe matter, we only need to include a subset of the SM fields (restricting ourselves to the first generation):

- **Left-handed quark doublet**: $Q_L\left(\frac{1}{2}; \mathbf{3}, \mathbf{2}, +\frac{1}{3}\right)$

- **Right-handed up quark**: $u_R\left(\frac{1}{2}; \mathbf{3}, \mathbf{1}, +\frac{4}{3}\right)$

- **Right-handed down quark**: $d_R\left(\frac{1}{2}; \mathbf{3}, \mathbf{1}, -\frac{2}{3}\right)$

- **Left-handed lepton doublet**: $L_L\left(\frac{1}{2}; \mathbf{1}, \mathbf{2}, -1\right)$

- **Right-handed electron**: $e_R\left(\frac{1}{2}; \mathbf{1}, \mathbf{1}, -2\right)$

- **Higgs field**: $\phi(0; \mathbf{1}, \mathbf{2}, +1)$

We have here used the common names for these fields in the SM, but we do not restrict ourselves to interpreting them as SM fields. To identify each field, we have listed their

properties explained above in the following order:[4]

- **Spin** SO(1,3): $\frac{1}{2}$ for fermions, 0 for the Higgs boson. The helicity (left or right-handedness) of fermions is indicated by the subscript $L$ or $R$. This property is associated with the representation of the Lorentz group. In the following we will not include the spin label explicitly, but will state which fields are scalars and which are fermions. We will, however, include the $L$ or $R$ as part of the name of the field.

- The representation of **color** SU(3)$_C$

- The representation of **weak isospin** SU(2)$_L$

- The value of the U(1)$_Y$ **hypercharge** $Y$

The assignment of the above properties of the fields is not arbitrary. In the SM, these transformation rules are dictated by a combination of theoretical consistency and experimental observations that confirm the model's predictions. The symmetries of the theory have a direct connection to conservation laws and have measurable impacts on the interactions between particles.

An example of a gauge-invariant term in the Lagrangian is the Yukawa coupling for the up quark,

$$\mathcal{L}_Y = -y_u \bar{Q}_L \tilde{\phi} u_R + \text{h.c.} \tag{2}$$

Here, $\tilde{\phi} = i\sigma_2 \phi^*$ is the conjugate Higgs doublet, $\sigma_2$ is the second Pauli matrix, so that

$$i\sigma_2 = \begin{pmatrix} 0 & 1 \\ -1 & 0 \end{pmatrix}, \tag{3}$$

$y_u$ is the Yukawa coupling constant for the up quark, and "h.c." denotes the Hermitian conjugate.

This term is constructed to be invariant under the gauge transformations. Let's see how this works:

- **Color** SU(3)$_C$: $\overline{Q}_L$ transforms as $\bar{\mathbf{3}}$, $u_R$ as $\mathbf{3}$; The bar indicates that it transforms in the complex conjugate representation, a different 3-dimensional set of transformation rules. Their product transforms as a singlet, or in group theory language: $\bar{\mathbf{3}} \otimes \mathbf{3} \supset \mathbf{1}$

- **Weak Isospin** SU(2)$_L$: $\overline{Q}_L$ transforms as $\bar{\mathbf{2}} \sim \mathbf{2}$, $\tilde{\phi}$ as $\mathbf{2}$; their product also transforms as a singlet: $\mathbf{2} \otimes \mathbf{2} \supset \mathbf{1}$. The right-handed up-quark $u_R$ is already a singlet ($\mathbf{1}$) under SU(2)$_L$.

- **Hypercharge** U(1)$_Y$: The total hypercharge is zero:

$$Y_{\text{total}} = \left(-\frac{1}{3}\right) + (-1) + \left(\frac{4}{3}\right) = 0$$

We can encode all this information using indices for each group and contracting them appropriately. If we explicitly write them for Eq. 2 in two-component Weyl spinor notation,

$$\mathcal{L}_Y = -y_u Q^{\dagger\,i}_{L\,c\dot{\alpha}} \tilde{\phi}_i u_R^{c\dot{\alpha}} + \text{h.c.} \tag{4}$$

Here, the indices represent the components of each field corresponding to different symmetry groups:

---

[4]Note that for an antiparticle, the gauge symmetry representations are changed: for SU(3) we get the complex conjugate representation. For SU(2), the representations are (pseduo)real, so it does not change. For U(1), we have $Y \to -Y$.

- $c$ represent $SU(3)_C$ color indices;

- $i$ represent $SU(2)_L$ weak isospin indices;

- $\dot{\alpha}$ represent spinor indices in the dotted/undotted index convention (see [44] for the convention we use).

Repeated indices are contracted (i.e., summed over, $a^i b_i = \sum_i a^i b_i$), ensuring that the term is invariant under the symmetry transformations. Usually, these indices are not written explicitly but are implied.

## 2.4 Forces

Each fundamental force, and thus each associated symmetry group, comes with its own mediator particles, as follows:[5]

- The **strong force** is mediated by gluons, described by eight gluon fields $G_\mu^a$.

- The **weak force** is mediated by the $W^\pm$ and $Z$ bosons, described by the fields $W_\mu^\pm$ and $Z_\mu$.

- The **electromagnetic force** is mediated by the photon, described by the photon field $A_\mu$.

To construct a Lagrangian that includes these force-carrying particles and remains invariant under gauge transformations, we use the concept of the *covariant derivative $D_\mu$*, a modification of the usual derivative $\partial_\mu$ that accounts for the interactions introduced by the gauge fields. It allows us to differentiate fields in a way that respects the symmetry transformations of the theory. Essentially, it combines the ordinary derivative with the gauge fields, ensuring that the derivative of a field transforms in the same way as the field itself under gauge transformations.

In general, the covariant derivative is a complicated object. However, for illustrative purposes, consider the simplest case involving an Abelian symmetry (a U(1) symmetry where the order of transformations does not matter). In this scenario, for a particle field $\psi$ interacting with gauge fields, the covariant derivative is defined as:

$$D_\mu \psi = \partial_\mu \psi - ig A_\mu \psi \tag{5}$$

Here, $g$ is the coupling constant and $A_\mu$ represents the gauge fields associated with the Abelian symmetry group. The exact form of $D_\mu$ depends on the field's transformation properties and the gauge group involved. The corresponding formula for interactions governed by non-Abelian symmetries, i.e., the strong and weak forces in the SM, is a more involved generalization of the above equation, however, for our purposes this needs not be introduced explicitly.

## 2.5 Kinetic terms

An essential component of the Lagrangian in a QFT is the *kinetic term*, which describes how particles propagate through spacetime. For fermions like electrons and quarks, the kinetic term involves derivatives of the fields, capturing how these fields change from one point to another.

---

[5]Note that, in theories beyond the SM there might be additional gauge symmetries and, thus, additional force-carrying particles.

Using the covariant derivative, the kinetic term for a (Weyl) fermion field $\psi$ is written as

$$\mathcal{L}_{\text{kin}}^{\text{fermion}} = i\,\psi^\dagger\,\overline{\sigma}^\mu D_\mu \psi \tag{6}$$

where $\overline{\sigma}^\mu$ are the two-component Dirac matrices, and $\psi^\dagger$ is the hermitian conjugate of the field. Similarly for a scalar field $\phi$, the kinetic term is

$$\mathcal{L}_{\text{kin}}^{\text{scalar}} = D^\mu \phi^\dagger D_\mu \phi. \tag{7}$$

Thus, by using $D_\mu$ instead of $\partial_\mu$, we ensure that the kinetic term remains invariant under gauge transformations. This construction automatically includes the interactions with gauge bosons.

The kinetic term for gauge bosons can be constructed using the covariant derivative just like we did for matter fields. When we examine the commutator of two covariant derivatives acting on a field $\psi$:

$$[D_\mu, D_\nu]\psi = D_\mu D_\nu \psi - D_\nu D_\mu \psi \tag{8}$$

We find that this commutator involves terms that depend on the gauge fields themselves. This is because the covariant derivative includes the gauge fields in its definition. The commutator $[D_\mu, D_\nu]$ encapsulates how the gauge fields vary in spacetime and, importantly, how they interact with each other. The kinetic term for the gauge bosons can be encoded using the covariant derivative:

$$\mathcal{L}_{\text{gauge}} = -\frac{1}{2}\text{Tr}\left([D_\mu, D_\nu][D^\mu, D^\nu]\right) \tag{9}$$

This expression includes all the necessary terms for the gauge bosons to propagate and interact, while ensuring the Lagrangian remains invariant under gauge transformations. The trace is necessary because for non-abelian groups the covariant derivative is a matrix. Nonetheless, the important fact is that by using the covariant derivative, we can write the kinetic terms for both matter fields and gauge bosons in a way that respects the symmetries of the theory.

To summarize this long section in a brief slogan, *symmetries are described by groups, and quantum numbers are labels that describe how fields transform under symmetry transformations.* Individual fields do transform, but they are combined in the Lagrangian in ways that ensure the overall expression—the Lagrangian—*does not change.* This principle of building invariant terms is crucial for constructing consistent and physically meaningful theories and is encoded in the Lagrangian.

## 3 Training the model

We trained a model based on a Bidirectional and Auto-Regressive Transformer (BART) architecture [45], with approximately 357 million parameters, to generate Lagrangians when given a list of particle fields and their symmetries. We show the pipelines for data-generation and model inference in Figure 1. More technical details on the architecture and training process can be found in Appendix B. Below we describe the process for generating the datasets used to train the model in more detail.

### 3.1 The dataset generation pipeline

To generate training data, we built a pipeline that automates the process of generating Lagrangians from fields and symmetries, using a combination of AutoEFT [44] and our own

code. AutoEFT is a sophisticated tool that applies mathematical methods to find allowed interactions in effective field theories (EFTs). In this context EFTs refers to the formalism used to describe the effect of physics at energy scales significantly above the energy scale of the particles being studied at experiments. This works by adding extra terms to the Lagrangian. Conveniently, AutoEFT can also find the allowed Lagrangian terms for the interactions between a list of fields and their symmetries, even if no physics at a higher energy scale is present, which is what we use it for here. However, some of the terms (such as Kinetic and mass terms) are not calculated by AutoEFT, so we have added our own code to generate them.

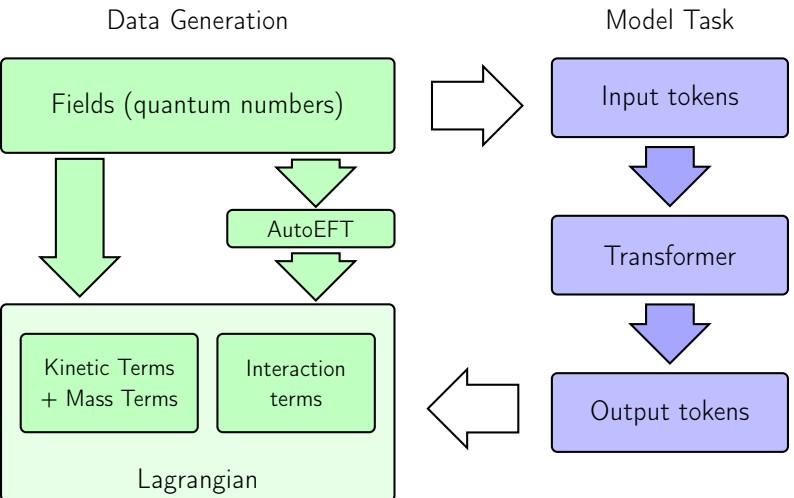

Figure 1: Data generation and model task. The left side shows the data generation pipeline, where fields and their quantum numbers are used to write a Lagrangian. The right side shows the model task, where the model takes input tokens and generates output tokens.

## 3.2 Sampling the space of Lagrangians

With our pipeline in place, we can generate any Lagrangian for a given set of fields and symmetries. However, the space of possible Lagrangians is vast, and we need to choose a representative subset to train our model.

While in principle we could allow for several gauge symmetries, we chose to focus on the $SU(3) \times SU(2) \times U(1)$ gauge group and its subgroups for simplicity and its relevance to the SM. This choice is easily extendable to more complex gauge groups, but provides a foundation for this initial work. It is important to note that the fields themselves do not need to be restricted to the SM fields. As we allow for fields that are singlets (do not transform) under one or more of the gauge groups, this also means that, e.g., a Lagrangian of a theory with only an $SU(2)$ symmetry can be generated as well. In a different view, one could also treat any of the groups as non-SM gauge groups, e.g., $SU(3)_R$ instead of $SU(3)_C$ ,$SU(2)_D$ instead of $SU(2)_L$, $U(1)_{L_\mu - L_\tau}$ instead of $U(1)_Y$.

We also restrict ourselves to only consider scalars and fermions as matter fields while vectors fields are restricted to gauge bosons. Gauge bosons are automatically considered whenever the corresponding gauge groups are required. Unless otherwise stated, we will use *fields* to only mean matter fields (which can only be scalars or fermions) for the remainder of this paper.

That leaves the question of how to choose the fields themselves. In other words, what Lagrangians should we include in a "*good*" dataset. The simplest approach would be to

generate $N$ random Lagrangians, as uniformly distributed as possible in the number and properties of the fields. We made such a dataset, which we refer to as "uniform dataset" for cross-validation purposes.

However, if we take inspiration from findings in training transformer models for natural language tasks, or even in symbolic expressions [33, 46, 47], there are significant gains when optimizing the dataset for learning. In our case, that means e.g. that we might want to oversample extreme or rare cases, as they might be interesting when it comes to the physics and they are also more likely to be challenging for the model. At the same time, what might be challenging for humans or for the actual implementation of the training, does not always align with what is challenging for the model. Take for example the fact that while longer Lagrangians, e.g. with more fields, might at first seem likely to be harder to learn, train set priming [47] suggests that the opposite might be true. By training the model on many shorter examples, it might only need a smaller sample of longer ones to match the performance of a model that was trained on a more uniform dataset. To test this hypothesis, we generated an optimized dataset (referred to as "sampled dataset") of ca. 280K Lagrangians with the following properties[6]:

**Field count distribution:** The number of fields in each Lagrangian was determined using a probability distribution that favors fewer fields. Specifically, there was a 25% chance each for Lagrangians to have one, two, or three fields, an 11% chance for four fields, and a 7% chance each for five or six fields. This approach resulted in a dataset rich in simpler examples, which are crucial for establishing foundational understanding in the model. It is worth noting that there are only 2997 one-field scenarios within the allowed quantum numbers. Oversampling one-field scenarios allows the model to learn them properly despite its small percentage to the full possible Lagrangian space. [7].

**Spin composition and field types:** Fields in the Lagrangians were randomly assigned spins of either 0 (scalars) or $\frac{1}{2}$ (fermions), with equal probability. Fermions are further assigned with helicities of either $-1/2$ (left) or $1/2$ (right) with equal probability. We observed that approximately 26% of the Lagrangians have a purely bosonic matter field content (containing only scalars), while about 21% are purely fermionic. The remaining Lagrangians contain both scalars and fermions in varying proportions.

**Gauge group representation:** In our dataset, for SU(3), fields could be in the triplet ($\mathbf{3}$), antitriplet ($\bar{\mathbf{3}}$), or singlet ($\mathbf{1}$) representations. For SU(2), fields could be in the singlet ($\mathbf{1}$), doublet ($\mathbf{2}$), or triplet ($\mathbf{3}$) representations. The $U(1)_Y$ hypercharges were assigned as random fractions with numerators ranging from $-9$ to $9$ and denominators from $1$ to $9$, which were then cast as simplified fractions (e.g. a draw of $3/9$ would be represented as $\frac{1}{3}$). [8]

**Enrichment of trilinear interactions:** Trilinear interactions are those that have three fields interacting with each other. Such an interaction corresponds to a term in the Lagrangian that involves the product of three fields. That means that the three fields have to be in a representation that allows for such a term to be invariant. In a uniformly sampled dataset, these interactions would be rare. However, we know that trilinear interactions are crucial in particle physics—that is, e.g., how the Higgs boson interacts with fermions—so we enriched our dataset with Lagrangians that contain trilinear interactions. This includes both scalar trilinear interactions (three scalars) and Yukawa interactions (a scalar with

---

[6]The data distribution described here is prior to applying a sequence length filter to account for the model's context length.

[7]This is similar to how using log-uniform distribution allows transformer to perform better at the greatest common divisor task [33]

[8]In the uniform dataset, fields have only positive U(1) hypercharges. This does not impact the generality of Lagrangians from a physics perspective, it is just a convention amounting to deciding what to call a field or its complex conjugate.

450 two fermions). During dataset generation, after the fields were assigned quantum numbers,
451 we adjusted the sampling strategy to preferentially include Lagrangians with trilinear
452 interactions so that approximately 50% of the Lagrangians with more than two fields
contained trilinear interactions. Figure 2 shows the training data distributions within the

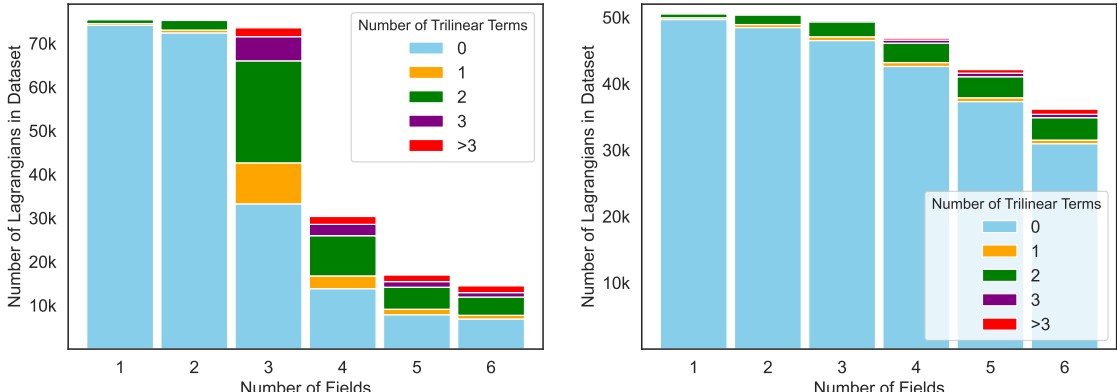

Figure 2: Training Data Distribution of both sampled (left) and uniform (right) datasets.

453
454 context length (2048 tokens) of our model. [9]

455 Sometimes, two or more invariant Lagrangian terms might describe the same physics,
456 which means that in principle one can only include one of them. As we are exploring the
457 transformer's capability to build invariant terms, we focused on all possible terms including
458 such redundant terms. Going towards non-redundant scenarios would add another layer
459 complexity for the model to overcome, which we leave for future work. Two transformer
460 models were trained based on the datasets: the sampled model (trained on the sampled
461 dataset) and the uniform model (trained on the uniform dataset).

462 ## 3.3 Encoding fields and Lagrangians

Table 1: Full vocabulary. Use cases are shown in Table 2 to Table 11.

| Token types | Tokens |
|---|---|
| Math | i, +, -, /, COMMUTATOR_A, COMMUTATOR_B |
| Numbers | 0, 1, 2, 3, 4, 5, 6, 7, 8, 9 |
| ID | ID0, ID1, ID2, ID3, ID4, ID5, ID6, ID7, ID8, ID9 |
| Field | FIELD, SPIN, HEL, DAGGER |
| Lorentz | DERIVATIVE, SIGMA_BAR |
| Symmetry | SU3, SU2, U1, LORENTZ, CONTRACTIONS |
| Misc. | SOS, EOS |

463 To enable effective learning of symbols and equations by the transformer model, a crucial
464 preprocessing step is the tokenization of fields and Lagrangians. As we see in Section 2,
465 Lagrangians are made up of mathematical objects with a wide range of symbols to represent
466 them – quantum fields ($\phi, \psi, A_\mu, Q_L, ...$), operators ($\bar{\sigma}^\mu, D_\mu, \dagger, +, -, /, ...$) and constants

---

[9]Following the tokenization method in Section 3.3, all Lagrangians are required to have less than 2048 tokens (see Appendix B). This decreases the number of Lagrangians with long sequences as seen in Figure 2.

Table 2: Example of object-tokenization.

| Object | Tokens |
|--------|--------|
| $\phi(\overline{\mathbf{3}}, \mathbf{2}, -1/3)$ | FIELD, SPIN, 0, SU3, -, 3, SU2, 2, U1, -, 1, /, 3, ID3 |
| $\psi_L(\overline{\mathbf{1}}, \mathbf{3}, 0)$ | FIELD, SPIN, 1, /, 2, SU2, 3, HEL, 1, /, 2, ID9 |
| $\phi^\dagger(\mathbf{1}, \mathbf{1}, 7/5)$ | FIELD, SPIN, 0, U1, 7, /, 5, DAGGER, ID7 |
| $D_\mu^{(\mathrm{SU}(2), \mathrm{U}(1))}$ | DERIVATIVE, SU2, U1, ID4 |
| $D_\mu^{(\mathrm{SU}(3), \mathrm{SU}(2), \mathrm{U}(1))}$ | DERIVATIVE, SU3, SU2, U1, ID1 |
| $\partial_\mu$ | DERIVATIVE, ID5 |
| $\overline{\sigma}^\mu$ | SIGMA_BAR, ID2 |
| $g^{\mu\nu}$ | LORENTZ, ID6, ID4 |
| $\epsilon^{ab}$ | SU2, ID0, ID7 |

$(m, i, \lambda, ...)$. [10] Objects in different symmetry representations will have corresponding indices, and an invariant term will have all the indices contracted (summed over) and whether they are explicitly written in the Lagrangians depends on the context in which the expression is being used. For instance, one could in principle write Lagrangians without any group representation indices and thus leave for the reader the task of figuring out how they should be contracted. This would not be very useful for understanding whether a model has learned to build invariant terms. On the other hand, too much information explicitly written would lead to intractably large expressions. In other words, care has to be taken to choose the right amount of verbosity in how the objects are represented.

Hence, we introduce a tokenization scheme that ensures that all relevant information is preserved and accessible to the transformer model with generalizability and interpretability in mind. Table 1 provides the full vocabulary of our tokenization method [11] and Table 2 provides example scenarios of object-tokenization. Full tokenization of various terms are provided in appendix C. In this paper, tokens are presented as strings in `teletype font` separated by commas.

Our tokenization scheme works as follows:

**Fields:** As discussed in Section 2.2, a field is defined by its quantum numbers, akin to how words are built from their alphabets. A dagger (†) represents the mathematical operation of Hermitian conjugation. The relevant information that uniquely identifies each field is included as separate tokens following the `FIELD` token.[12] A field is tokenized with the following order:

1. Field token             :    `[FIELD]`

2. Spin tokens             :    `[SPIN,0]` or `[SPIN,1,/,2]`

3. Symmetry tokens (if required) `[SU3,3]`,`[SU2,3]`,`[U1,2]`,`[U1,-,4,/,7]`, ...

---

[10] As certain mathematical objects can be made up of one or more symbols, we will use the words *objects* and *symbols* interchangeably in this work.

[11] Byte Pair Encoding (BPE) [48], is not used to allow better interpretability and generalizability. BPE does not work well with Out-Of-Vocabulary (OOV) words in translation tasks. [49]. Vocabulary of subwords learned during training may not be optimal when domain-specific terms or rare words arises. In our case, this could lead to failure modes for the model when dealing with OOV fields. BPE is also a frequency-based method rather than context based. This will allow cases such as "3 SU2" to be a possible *subword* token that provides no context, hindering us from interpreting what happens in our transformer model, which is done via embedding analysis in Section 6.

[12] An alternative would be to pre-assign symbols to all possible quantum fields within our dataset, but this would limit the flexibility of the tokenization method and its potential for generalization.

4. Helicity tokens (if required):  `[HEL,1,/,2]` or `[HEL,-,1,/,2]`

5. Dagger token (if required) :  `[DAGGER]`

6. ID token  :  `[ID4]`,`[ID2]`,`[ID0]`, ...

Examples are shown in the first three rows of Table 2. ID tokens are randomly assigned to a field for every term in the Lagrangian, serving as indices for the purpose of keeping track of contractions, since the specific names of contracted indices are not relevant. ID and dagger tokens are not used in input fields. We represent fermion fields following the two-component spinor formalism. As we will see next, gauge bosons are not explicitly represented as spin-1 vector fields, but rather implicitly encoded within the covariant derivatives.

**Derivatives:** By writing all gauge interactions through covariant derivatives, we keep the Lagrangians more compact after tokenization while still preserving the full physics content. In particular, the covariant derivatives implicitly contain the gauge fields, allowing us to recover the explicit gauge interactions simply by expanding them out. Although at first glance this may seem too compact, the important details remain encoded in the Lagrangian. Explicitly listing every gauge field interaction would lengthen the Lagrangians considerably and is not strictly necessary for our main goal, which is to see whether the model can learn to build invariant terms with scalars and fermions, as well as recognize where gauge fields should be included. This approach represents a trade-off that reduces token count without sacrificing essential physics. As such, derivatives are tokenized with the following order:

1. Derivative token  :  `[DERIVATIVE]`

2. Symmetries tokens (if required)`[SU3]`,`[SU2]`,`[U1]`,`[SU2, U1]`,`[SU3, SU2, U1]`,...

3. ID token  :  `[ID7]`,`[ID2]`,`[ID1]`, ...

Examples are shown in the forth and fifth rows of Table 2.

**Commutators:** Following the convention of writing the gauge interactions with covariant derivatives, we represent the kinetic terms for gauge bosons as shown in equation 9. This choice requires us to encode commutators. We do so in the following way:

$$[X, Y] = \texttt{COMMUTATOR\_A}, X, \texttt{COMMUTATOR\_B}, Y \tag{10}$$

where "`COMMUTATOR_A`" and "`COMMUTATOR_B`" are positional tokens that indicate positions within the commutators while X and Y can be any arbitrary tokens. Traces ("`Tr`") are left implicit based on the covariant derivatives in the commutator. An example use case is shown in Table 9 of Appendix C.

**Constants and Parameters:** Given that our focus is on the symmetry conservation of Lagrangians, we do not encode any constants or parameters [13]. Symbolically, couplings and masses are constants that are present in every interaction term and mass term, respectively, but provide no extra information regarding symmetry. Their numerical values are also typically continuous and inferred experimentally.

**Contractions:** In this tokenization scheme, each Lagrangian term consists of two parts: the tokenized objects themselves and the accompanying contraction information, which appears after the "`CONTRACTION`" token. The contraction information specifies how the symbols in each term are contracted—something that a human physicist interprets intuitively but that a transformer model cannot infer from the fully contracted Lagrangian alone. Conceptually, it serves the same purpose as specifying the Minkowski metric for Lorentz indices or the Levi-Civita symbol for other indices. When necessary, this contraction information is provided in the following format:

---

[13]Except for the imaginary unit, "`i`" which has its importance in the kinetic term of fermions

1. Contraction token                                         :[CONTRACTION]

2. Symmetries tokens indicating indices type :[LORENTZ] or [SU3] or [SU2]

3. ID token of contracted indices                 :[ID7, ID2],[ID1, ID1, ID4], ...

4. Repeat 2. and 3. until all contractions are done

Examples of contraction types are shown in the last two rows of Table 2. Full terms with contraction information can be found in Appendix C.

In our dataset, we do not include scenarios with repeating fields, nor repeating symmetry groups. However, our tokenization method allows to extend to these scenarios by simply introducing extra tokens to name different generations (flavors) and groups accordingly. Generalization towards more complicated groups can also be done via the use of Dynkin labels for the representations instead of naming them by their dimension.

# 4   Performance on test datasets

In this section, we evaluate the performance of the models on the task of predicting Lagrangians. To evaluate our models' performance and to compare them, we generated three datasets. First, two new datasets of around 32,000 Lagrangians each, one for each sampling strategy explained in Section 3.2. We removed any Lagrangian appearing in the original training datasets, leaving only Lagrangians that the models never saw during training (or evaluation during training). The third test dataset was made by merging both datasets leaving us with around 64,000 Lagrangians.

For training the model, it was enough, as we will see, to use cross-entropy loss as an estimation of how close the model's prediction is to the actual Lagrangian. But to evaluate the actual performance of the model, we need to account for the fact that the order of terms in the Lagrangian does not matter, and that the order of appearance of fields and symbols does not matter [14] either as long as the correct contractions are made, e.g. $\epsilon^{abc}\phi_a\phi_b\phi_c = \epsilon^{abc}\phi_b\phi_c\phi_a$.

That means that to assess the performance of the models, we will need to introduce several error metrics. Given a Lagrangian with $N_{\text{Expected}}$ terms and $N_{\text{Predicted}}$ terms predicted by the transformer model, we define the following:

$$\text{Object Score, } S_{\text{Object}} = \frac{N_{\text{Correct Objects}}}{N_{\text{Expected}}} \tag{11}$$

where $N_{\text{Correct Objects}}$ is the number of predicted terms that correctly match the expected terms in objects (fields, derivatives, etc.) without caring for contractions.

$$\text{Contraction Score, } S_{\text{Contraction}} = \frac{N_{\text{Correct Contractions}}}{N_{\text{Expected}}} \tag{12}$$

where $N_{\text{Correct Contractions}}$ is the number of predicted terms that correctly match the expected terms in both the objects (fields, derivatives, etc.) and contractions.

$$\text{Length Penalty, } P_{\text{Length}} = \frac{N_{\text{Extra}}}{N_{\text{Expected}}} \tag{13}$$

---

[14]When dealing with fermionic fields, this statement is true up to a potential minus sign. Given that we do not care about numerical values of couplings of constants, for the purpose of this paper this is not important.

568 where $N_{\text{Extra}}$ is the total number of extra terms in the predicted Lagrangian (ie. $N_{\text{Predicted}} - N_{\text{Expected}}$
569 when $N_{\text{Predicted}} > N_{\text{Expected}}$ ).

$$\text{Lagrangian Score,} \ S_{\text{Lagrangian}} = S_{\text{Contraction}} - P_{\text{Length}} \qquad (14)$$

570

571 The Lagrangian score quantifies how accurately the predicted Lagrangian matches the
572 expected Lagrangian, considering both the correctness of individual terms and the total
573 number of terms. A perfectly predicted Lagrangian would have a Lagrangian score of 1
574 with a contraction score of 1 and a length penalty of 0. The length penalty penalizes any
575 extra terms in the predicted Lagrangian while the contraction score penalizes both missing
576 and wrong terms. It is also worth noting that $S_{\text{Contraction}}$ can only be less than or equal to
577 $S_{\text{Object}}$, since the objects need to be correct for the contraction to be considered.

Table 3: Overall accuracy on the test dataset with respect to different metrics.

| Metric | $S_{\text{Lagrangian}} = 1$ | | $S_{\text{Contraction}} = 1$ | | $S_{\text{Object}} = 1$ | | $P_{\text{Length}} = 0$ | |
|---|---|---|---|---|---|---|---|---|
| Models | Sampled | Uniform | Sampled | Uniform | Sampled | Uniform | Sampled | Uniform |
| Sampled dataset | **93.4%** | 89.9% | **95.8 %** | 90.2% | **95.9%** | 90.5% | 97.4% | **99.6%** |
| Uniform dataset | 90.5% | **96.6%** | **96.9 %** | 96.9% | 97.0% | 97.0% | 93.4% | **99.7%** |
| Merged dataset | 92.0% | **93.2%** | **96.3 %** | 93.5% | **96.4%** | 93.7% | 95.5% | **99.7%** |

578      Table 3 shows the accuracies of both models across different metrics and datasets.
579      Overall, both models performed well on the merged dataset, with 92% of predictions
580 having a perfect Lagrangian score for the sampled model and 93.2% for the uniform model.
581 Each model works relatively well on their own dataset while the uniform model excels in
582 minimizing length penalization. The sampled model achieves similar object and contraction
583 scores for the uniform dataset but the same cannot be said for the uniform model on the
584 sampled dataset.
585      The merged dataset provides the overall trend of models: The sampled model performs
586 well in object and contraction scores (with less than 5% errors), while the uniform model
587 performs slightly better in length (with less than 1% having extra terms). This is likely due
588 to the higher fraction of trilinear interactions in the sampled training data, i.e. the sampled
589 model saw more complex terms but also longer Lagrangians on average. Conversely, the
590 uniform model better avoids adding spurious terms, reflecting the more balanced coverage
591 of sequence lengths in its training data.
592      These effects are small, indicating that both approaches robustly train the model to
593 predict valid Lagrangians across a wide range of symmetry and field configurations. Figure 3
594 shows the distribution of the Lagrangian scores on the test dataset for both models. The
595 negative scores are due to the penalty for extra terms in the predicted Lagrangian.
596      To provide more insight, we look at the cumulative distribution of the fraction of
597 incorrect terms in the predicted Lagrangians ($1 - S_{\text{Correct}} = N_{\text{Wrong}}/N_{\text{Expected}}$) on the
598 merged test dataset. Figure 4 and Figure 5 show the cumulative distributions separated by
599 the number of fields and presence of trilinear interactions in the Lagrangians, respectively.
600 Here, we can see that the sampled models consistently perform better than or comparable
601 to the uniform model across different $n$-field scenarios. However, while both models perform
602 exceptional well ($>97\%$) for Lagrangians without trilinear interactions, the sampled model
603 works considerably well for the Lagrangians with trilinear interactions compared to the
604 uniform model.
605      In almost every case (except for the 6-field scenario by the sampled model), close to 99%
606 of the predicted Lagrangians have less than 20% errors. Despite only 7% of training data

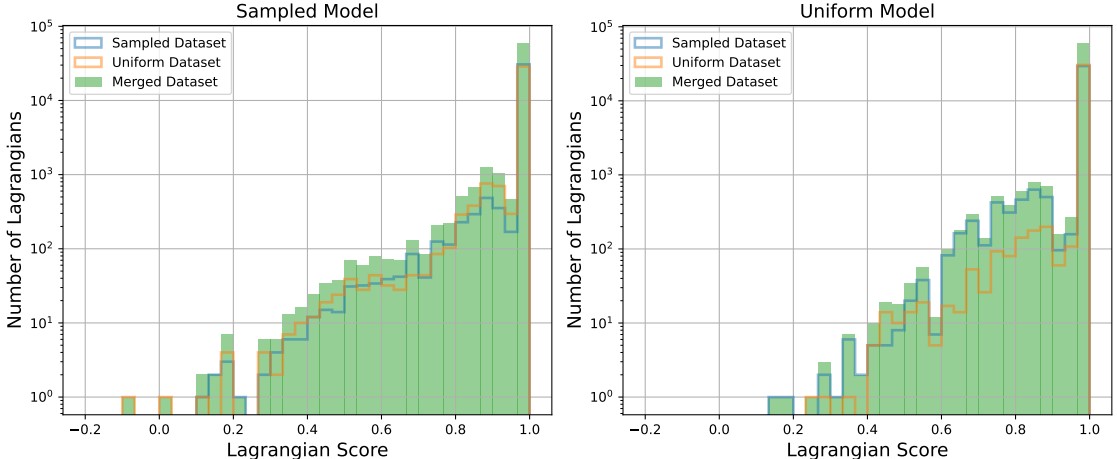

Figure 3: Distribution of the Lagrangian scores (Eq. 14) for the sampled model (left) and the uniform model (right).

being 6-field examples, the sampled model still handles these more complex cases remarkably well and comparable to the uniform model (trained with 16% of training data containing six-field examples). This outcome is encouraging, as it demonstrates that focusing on shorter sequences during training can yield a model capable of effectively handling longer inputs without significantly compromising performance. Further improvement on the dataset would then require to balance the amount of long sequences and the amount of Lagrangians with trilinear terms, such that both a minimal priming rate is obtained and the model does not overestimate the number of terms.

It is also worth noting that, even on expected Lagrangians from the training dataset, the model generates Lagrangians with ID tokens that differ from the expected Lagrangians. This strongly suggests that the model has learned the concept of dummy indices ($\epsilon^{ab}\phi_a\phi_b = \epsilon^{cd}\phi_c\phi_d$). Example cases such as SM and Beyond the Standard Model (BSM)

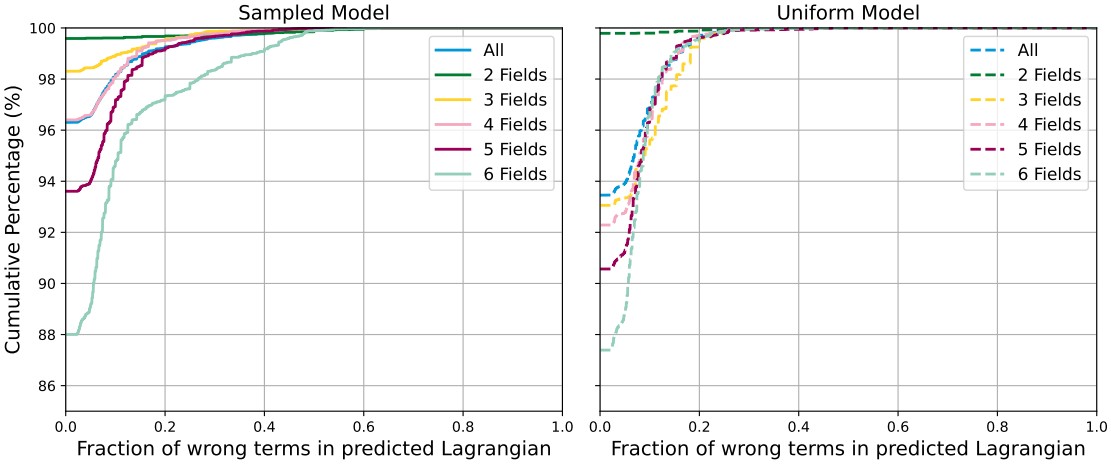

Figure 4: Cumulative distribution of the fraction of wrong terms in predicted Lagrangian with different number of fields. On the left (solid lines) is the sampled model, and on the right (dashed lines) is the uniform model.

scenarios are also examined in more details in Appendix D.

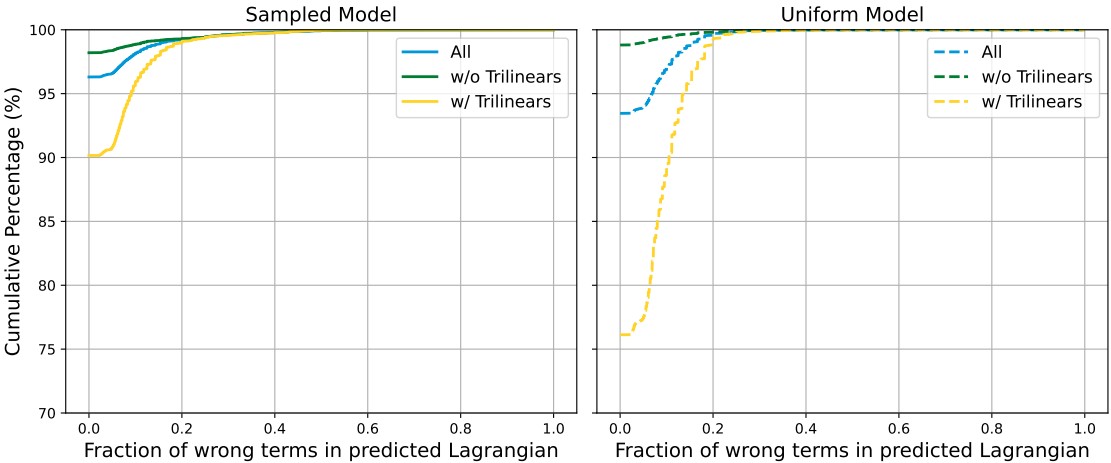

Figure 5: Cumulative distribution of the fraction of wrong terms in predicted Lagrangian with and without trilinear interactions. On the left (solid lines) is the sampled model, and on the right (dashed lines) is the uniform model.

# 5  OOD generalization

Thus far, our evaluation has focused on In-Distribution (InD) scenarios (scenarios with properties and qualities present in our datasets), where the obvious path towards improvements is to extend the training datasets and train a new model.

However, recall that our broader goal is to develop models capable of scaling into foundation models, and that requires models that are also able to conceptually generalize beyond the specific examples or the specific qualities of the examples seen during training.

Hence, it is equally important to analyze performance in the context of OOD generalization — how well a model deals with data that come from a distribution different (and with different qualities) from the training data. Examining OOD scenarios reveals why models fail under unfamiliar conditions and may inform the necessary next steps for improving a given architecture. Successful OOD generalization also serves as evidence of where the models rely on learned complex patterns rather than memorization, which is what enables the extrapolation to unseen cases.

We investigate two OOD scenarios, the capability of the model to generalize towards higher numbers of input fields, and unseen U(1) hypercharge. We will leave the details of the U(1) hypercharge case to Appendix E and focus for now on the higher number of input fields.

Our evaluation process involves assessing whether the generated Lagrangians are reasonable and U(1) conserving, testing the limits of OOD generalization and analyzing the nature of the failure modes. In our case, we consider a Lagrangian to be reasonable if:

- **Symbols contain the right amount of information**. E.g. a $\bar{\sigma}_\mu$ symbol should not have an SU3 token.

- **Fields contain reasonable quantum numbers**. SU(2) representation cannot be fractions. U(1) charges are either single digit integers or fractions with single digit integers as numerator and denominator, etc.

- **Contracted indices must come from the objects in the term**. The contraction information part of the term should not contain ID tokens that do not exist in the term.

- **Mass dimension of each term must be integers**. E.g. a term cannot contain only one fermion and one scalar.

- **Within Context Length**. The Lagrangian should be complete, so not reaching context length before "`[EOS]`"

- **Commutators should be paired**. Each "`[`" is closed by a "`]`".

- **No hallucination cases**. E.g. terms with a long chain of tokens in a nonsensical order.

## 5.1 Higher number of input fields

To evaluate the model's performance on higher number of input fields, we consider scenarios with up to 10 fields (going beyond the 6-field limit in the training dataset). To study this, 1K Lagrangians were generated using the sampled distribution for each $n$-field scenario, with the condition that none were present in the training data.

Table 4: Percentage of Reasonable Lagrangians and Lagrangians that conserved U(1)

| Data type | $N_{\textbf{Input Fields}}$ | Percentage of predicted Lagrangian (%) | | | |
|---|---|---|---|---|---|
| | | Reasonable | | U(1) conservation | |
| In-Distribution | 2 | 100.0 | | 100.0 | |
| | 3 | 99.98 | | 98.7 | |
| | 4 | 99.94 | 99.9 | 95.3 | 95.8 |
| | 5 | 99.8 | | 93.7 | |
| | 6 | 99.7 | | 91.4 | |
| Out-Of-Distribution | 7 | 99.6 | | 89.5 | |
| | 8 | 99.3 | | 86.6 | |
| | 9 | 99.3 | 99.5 | 87.2 | 87.5 |
| | 10 | 99.5 | | 86.8 | |

Table 4 shows the percentage of reasonable Lagrangians and U(1) conserving Lagrangians for each scenario. We can see that in every case, more than 99% of the predicted Lagrangians are reasonable. This is true even when we go towards OOD number of fields. This is a clear indication that it is able to extrapolate to OOD scenarios reasonably and not just produce gibberish. However, while still maintaining above 87% performance, there is a steady decrease in predicted Lagrangians that conserve U(1) symmetry as it goes towards more OOD scenarios.

Figure 6 shows the performance of the sampled model across different $n$-field scenarios. There is a steady decrease in the performance as the number of input fields grows, with relatively high performance still on 7-field scenarios (over 50% having at least Lagrangian score of 0.95). The spread of the score also increases with the number of fields, indicating that the performance is highly Lagrangian-dependent. To investigate failure modes, we look at the mean values of different metrics. While the length penalties are very low on average, we can see a similar decrease in the contraction score as in the Lagrangian score. If we look at the ratio of correct terms over predicted terms, we see that on average, at least 70% of predicted terms are correct even in the 10-field scenario. This hints that the model is mainly missing terms rather than consistently predicting the wrong terms.

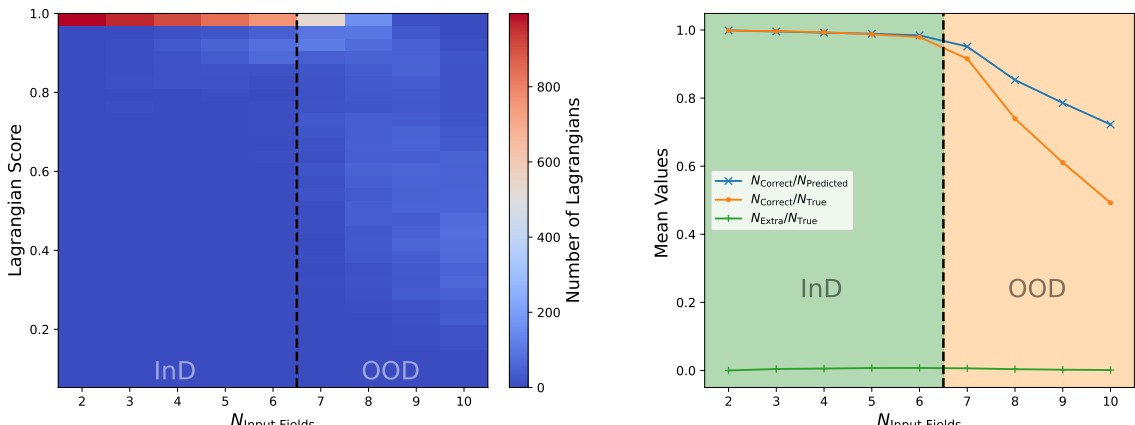

Figure 6: Lagrangian scores distribution (**left**) and mean values of different metrics (**right**) across different $n$-field scenarios. Dotted lines indicate the separation between InD dataset and OOD dataset. In the right plot, the metrics shown include the correct score($N_{\text{Correct}}/N_{\text{True}}$), length penalty($N_{\text{Extra}}/N_{\text{True}}$) and $N_{\text{Correct}}/N_{\text{Predicted}}$. Results are based on sampled model.

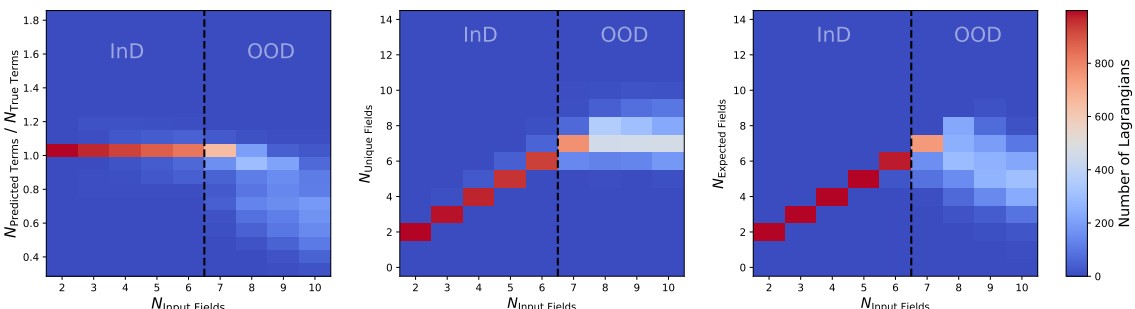

Figure 7: Failure modes of sampled model in OOD $n$-field scenarios. **Left**: Fractions of number of predicted terms over number of expected terms. **Middle**: The number of unique fields found in the predicted Lagrangians. **Right:** The number of expected fields used in predicted Lagrangians

Figure 7 illustrates the fractions of predicted terms, the number of unique fields, and the number of expected fields in predicted Lagrangians as functions of input fields, highlighting the model's failure modes. The first plot shows that the model misses more terms as the number of input fields increases. This can be attributed to the model's inability to count and keep track of input fields in OOD $n$-field scenarios. The BART architecture is built out of a non-causal encoder (encodes the information in the input fields) and a causal decoder (generates the Lagrangian). Non-causal Transformers, like BART's bidirectional encoder, are known to struggle with contextual counting tasks[15] [50]. The second plot reveals that while the model recognizes the need for more fields in OOD scenarios, it fails to handle more than 7-8 fields, struggling to count the "`FIELD`" tokens in the input. The third plot further indicates that the model fails to use expected fields in the prediction. The model tends to mix quantum numbers of different fields due to its inability to isolate regions of interest separated by "`FIELD`" tokens. These issues worsen with more input fields, emphasizing the role of contextual counting in such tasks. Despite these challenges, it is worth noting that BART's causal decoder (which better keeps track of counting e.g. how many `FIELD` tokens appear in each Lagrangian term) ensures that reasonable Lagrangians can still be written

---

[15]That is, identifying a specific region of interest within a sequence and perform accurate counting.

in OOD scenarios with almost 100% of OOD scenarios have all terms with reasonable mass dimension.

# 6 Embedding analysis

In this section, we shift from evaluating the model's performance to interpreting (as much as possible) what exactly it has learned. Specifically, we want to check whether the model has captured the concept of symmetry through its embeddings of the input sequences or not. To extract the learned representations, the desired inputs are encoded through the model and the embedding vectors (the output of the encoder's final layer when encoding the SOS token during the generation of the first token) are obtained. Again, only the sampled-model is used here.

## 6.1 Symmetry clusters

We begin by examining how individual fields are represented in the embedding space. To analyze these embeddings, we apply t-SNE (t-distributed Stochastic Neighbor Embedding) to reduce the dimensionality of the embedded vectors from 1024 to 3 dimensions. The resulting visualization is shown in Figure 8, illustrating how the fields are represented based on different group representations.

In terms of Lorentz representation, we observe distinct clusters that separates scalars and fermions. Notably, fermions form a single cluster but are separated into left-handed and right-handed components within that cluster. In the context of SU(3) and SU(2) groups, singlet fields tend to occupy a central region within the t-SNE space, while gauged fields are distributed towards the sides. For U(1) symmetry, integer representations gravitate towards the center of the clusters, while fractional representations appear on the shell. This differentiation highlights the structure and relationships between various field types and their symmetries being learned by the transformer. It is worth noting that while the model is tasked only to generate Lagrangians, it constructed different symmetry clusters in its embedding space.

## 6.2 Conjugation axis

Given that the model has learned some aspects of symmetry, we now investigate whether it has captured the more abstract concept of conjugation. In quantum field theory, "conjugation" generally refers to transforming a field into another with exactly opposite quantum numbers (its "conjugate"). While both the field and its conjugate appear in the Lagrangian, only one is explicitly listed among the inputs, and it is largely a matter of convention which one is labeled as "field" versus "conjugate".

Conjugation operations on quantum fields can be thought in the same way as certain semantic relationships in language models. For example, in natural language processing (NLP), relationships like gender or tense between words correspond to consistent vector offsets (or directions) in the embedding space (see Appendix F for details). Similarly, we assess whether consistent vector differences (offsets) exist between the embeddings of fields and their conjugates, which would indicate that the model represents conjugation as a consistent transformation in the embedding space. An illustration of this concept is shown in Figure 9.

Our analysis reveals a distinct axis of preference for fermion conjugation within the embedding space. We evaluated this by computing the absolute cosine similarity, $|s|$, between vectors $\vec{v}_{C_i}$ which are the difference between fields and their conjugation, i.e.

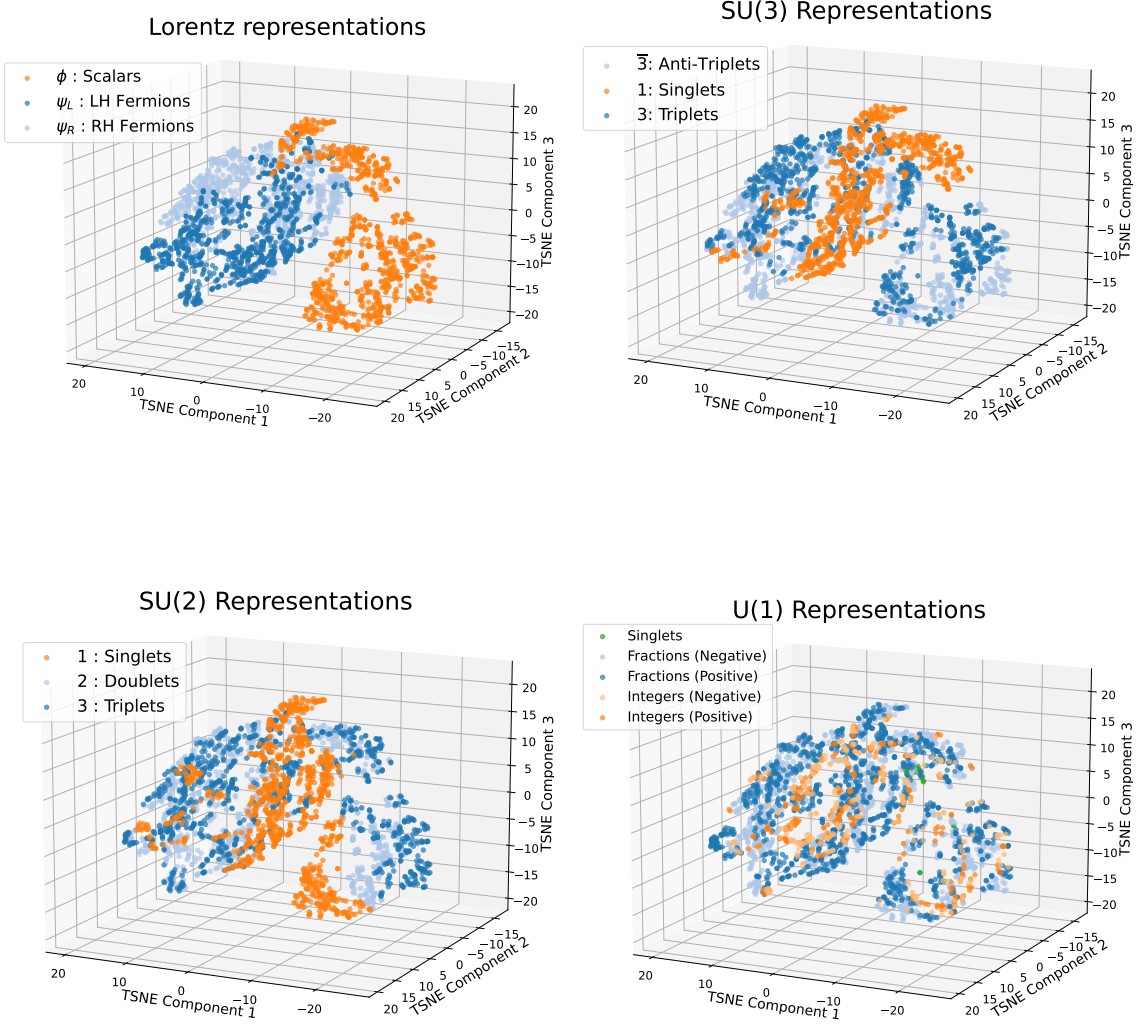

Figure 8: t-SNE visualization of individual field embeddings. Each plot shows the representation of the field according to their symmetry group.

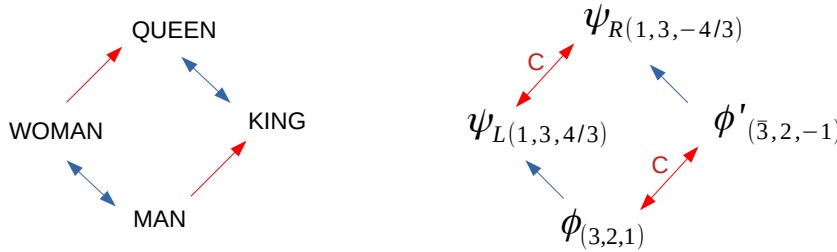

Figure 9: **Left**: Vector offsets of word pairs illustrating the gender and royalty relation. **Right**: Vector offsets of field pairs illustrating the conjugation relation

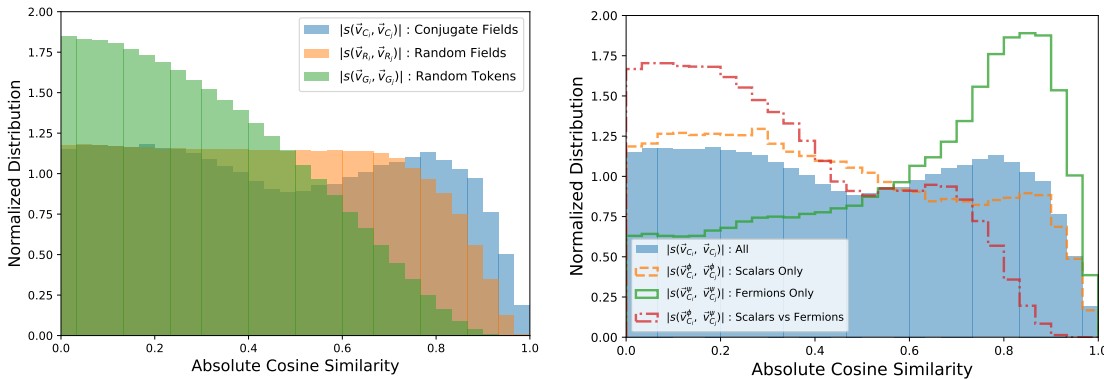

Figure 10: **Left**: Normalized distribution of absolute cosine similarities, between pairs of conjugation vectors, pairs of random translation vectors and pairs of random transformation vectors. **Right**: Normalized distribution of absolute cosine similarities between pairs of conjugation vectors separated according to the spins of the fields.

$\vec{v}_{C_i} = \vec{v}_{F_i} - \vec{v}_{F_i^C}$.[16] As shown in Figure 10 (Right), the distribution for fermion pairs peaks significantly near $|s| = 1$, indicating that the model assigned a specific axis in embedding space to the concept of conjugation.

To place this finding in context, we established two baseline comparisons as shown in Figure 10 (Left). The first, termed "Random Fields", involves calculating $|s|$ between vector differences of randomly paired field embeddings *fields* ($\vec{v}_{R_i} = \vec{v}_{F_a} - \vec{v}_{F_b}$). The resulting distribution shows a broad plateau, as expected given that the vectors represent embeddings of fields rather than random embeddings. The second baseline, "Random Tokens", uses differences between embeddings of randomly generated token sequences ($\vec{v}_{G_i} = \vec{v}_a - \vec{v}_b$).[17] The $|s|$ distribution for these pairs peaks sharply at 0, reflecting the expected lack of inherent directional structure in random sequences.

Compared to these baselines, the overall distribution for conjugation vectors peaks near $|s| = 0$ and $|s| \approx 0.8$. When further separated by spin (see Figure 10, right), it becomes evident that this axis is most strongly associated with fermions. Scalar fields, by contrast, do not exhibit as strong axis of preference. This difference arises partly because the relation between conjugation axes of different symmetry groups and the additional helicity conjugation that fermions have. For a more detailed discussion, please refer to Appendix F.2.

All this suggests the model successfully learned the concept of conjugation, particularly identifying a specific axis for fermions. It is worth noting that there are significant portions of the distributions' density away from 1 (i.e. exact relationship). Since every field can participate in different relationships other than conjugation (such as specific gauge transformations, U(1) charge differences), it is expected that the cosine similarity values are not exactly 1 for all pairs. In fact, it is beneficial for the model to represent multiple relationships, allowing it to capture a richer structure in the embedding space. This is consistent with the results in language models, where the vector offsets for different relationships are not always perfectly aligned [51].

---

[16]where s = +1 for two proportional vectors, s = 0 for orthogonal vectors, and s = -1 for opposite vectors. See Appendix F for further details

[17]These sequences are generated randomly with tokens from the vocabulary with varying length up to the maximum number of tokens that field could have, not including the SOS and EOS tokens.

# 7  Accessing model and datasets

We have made available the trained models and training datasets to the community. The trained models can be easily used to generate Lagrangians using the Hugging Face Transformers library [52].

The model is available at:
    https://huggingface.co/JoseEliel/BART-Lagrangian.

The training datasets are available at:
    https://huggingface.co/datasets/JoseEliel/lagrangian_generation.

An interactive example for demonstration purposes is available at:
    https://huggingface.co/spaces/JoseEliel/generate-lagrangians.

This example allows users to input a set of fields and the model will generate a Lagrangian based on the input, with a simple LaTeX output. We however recommend that users download the model and run it locally for more complex tasks.

# 8  Conclusions and outlook

We have shown that, given a list of particle fields, transformer models are capable of writing the corresponding particle physics Lagrangian, i.e., information-loaded equations with many symmetries involved that express the fundamental workings of the subatomic world. We successfully trained a modestly large transformer on such a task, which required a specialized dataset and a carefully thought out tokenization scheme.

The model has shown to be capable of writing Lagrangians with high accuracy both for randomly picked lists of fields but also for examples taken from actual physical models describing nature. The performance has a clear dependence upon the data distribution, mainly on the type of terms (taking care of special cases) and the number of fields (sequence length). Nonetheless, we observe that certain OOD scenarios—especially those featuring more than six input fields or unusual hypercharges—lead to diminished accuracy, indicating a need to refine how the model "counts" tokens and how it interprets numbers to maintain a correct field-by-field usage.

Our evaluations further reveal specific failure modes: for instance, the model may miss terms when many fields are required (e.g., in eight-field scenarios) or incorrectly contract indices in trilinear interactions under unusual conditions.

Notably, despite the task being Lagrangian generation, our evaluation also shows that the concepts of symmetry groups and representations, conjugation operation and even the use of dummy indices, have all been learned by the model. This suggests that the underlying mechanism is robust but that specialized counting architecture or data rebalancing could help mitigate the issue of missed or faulty terms, especially when large numbers of fields.

Given the general approach in our methods, extension of current work can be easily adapted, such as complicated symmetry groups, flavor generations, discrete symmetries, vector fields, higher mass dimensions (i.e., EFT terms), and much more. Future work will investigate adjusted transformer architectures or enhanced datasets to tackle these limitations, aiming to scale the approach toward building a broader "foundation model" for particle physics, capable of both precise symbolic manipulation and data-driven discovery.

Our work here lays one of the fundamental steps towards that goal. With many existing

efforts on using ML methods to extract as much information from experimental data as possible, a ML connection towards particle physics theory grows increasingly crucial. By having such a model, one can proceed in many directions: from experimental data directly to theoretical models, from theoretical models directly to simulated predictions and experimental data, and from equations to equations or data to data. These improvements— especially around OOD performance—will help ensure stronger reliability and expand the practical reach of these theoretical models in guiding future experiments and discoveries.

The future direction of this endeavor is very ambitious. In present-day particle physics, there is an expectation of some BSM scenario being realized in Nature in order to solve the problems of the current description (the SM). From both a theoretical and an experimental perspective our vision is to eventually input into the ML environment aimed at constructing *theoretical* Lagrangians also *experimental* information, in the form of real data showing deviations from the SM predictions, which could manifest as a resonance, a threshold effect, a Jacobian peak, etc. Ultimately, such a dataset will contribute to training ML models to help physicists select Lagrangians that are able to make predictions matching such data, through a truly agnostic approach to new physics dispensing of any (human) prejudice, which inevitably privileges always one or another realization of it.

# Acknowledgements

YSK would also like to thank François Charton for helpful discussions regarding data preparation and Thorsten Glüsenkamp regarding transformer architectures. ECM and YSK would like to thank Stefan Leupold for insightful physics discussions.

**Author contributions**   This project started from ECM's initial idea combined with YSK's insights for a feasible technical implementation. Following joint discussions on specifications, YSK generated the datasets. ECM managed the training, while YSK set up the analysis pipeline. Joint discussions between ECM and YSK informed the analysis results. RE provided guidance on group theory and SM on particle phenomenology. All authors participated in discussions throughout the project and contributed to writing the manuscript, which was approved by all.

**Funding information**   SM is supported in part through the NExT Institute and the STFC Consolidated Grant ST/X000583/1. The models were trained on Google Cloud Platform using research credits awarded to this project. The computations and data handling were enabled by resources provided by the National Academic Infrastructure for Supercomputing in Sweden (NAISS), partially funded by the Swedish Research Council through grant agreement no. 2022-06725. Specifically, we thank Chalmers e-Commons at Chalmers and the PDC Center for High Performance Computing at KTH Royal Institute of Technology, Sweden for providing access to the computing resources and data storage used in this research.

## A  Gauge group singlets

Triplets under $SU(3)_C$ can be combined in several different ways to obtain a singlet: reducing tensor products into irreducible representations, we have $\mathbf{3} \otimes \bar{\mathbf{3}} = \mathbf{1} \oplus \mathbf{8}$, meaning that it is possible to construct a singlet out of a $\mathbf{3}$ and a $\bar{\mathbf{3}}$, so this is an allowed combination. On the other hand, $\mathbf{3} \otimes \mathbf{3} = \bar{\mathbf{3}} \oplus \mathbf{6}$ which does not contain a singlet, so we cannot combine two triplets into a singlet. Adding one more triplet works, since $\mathbf{3} \otimes \mathbf{3} \otimes \mathbf{3} = \mathbf{1} \oplus \mathbf{8} \oplus \mathbf{8} \oplus \mathbf{10}$.

For $SU(2)_L$, we have the following combinations that involve two fields:

$\mathbf{2} \otimes \mathbf{2} = \mathbf{1} \oplus \mathbf{3}$

$\mathbf{3} \otimes \mathbf{3} = \mathbf{1} \oplus \mathbf{3} \oplus \mathbf{5}$

$\mathbf{2} \otimes \mathbf{3} = \mathbf{2} \oplus \mathbf{4}$

Thus, only the first two are viable terms in the Lagrangian, while a $\mathbf{2}$ and a $\mathbf{3}$ cannot form a singlet. Thus in kinetic and mass terms, two doublets or two triplets must be combined. For three fields we have

$\mathbf{2} \otimes \mathbf{2} \otimes \mathbf{3} = \mathbf{1} \oplus \mathbf{3} \oplus \mathbf{3} \oplus \mathbf{5}$

$\mathbf{3} \otimes \mathbf{3} \otimes \mathbf{3} = \mathbf{1} \oplus \mathbf{3} \oplus \mathbf{3} \oplus \mathbf{3} \oplus \mathbf{5} \oplus \mathbf{5} \oplus \mathbf{7}$

$\mathbf{2} \otimes \mathbf{3} \otimes \mathbf{3} = \mathbf{2} \oplus \mathbf{2} \oplus \mathbf{4} \oplus \mathbf{4} \oplus \mathbf{6}$

This means that we the first two combinations can indeed produce singlets under the $SU(2)_L$ gauge group, while the last one cannot. There are also allowed and disallowed combinations of four fields, but we will not display them here.

## B  Technical details of the model

We implemented a model with a BART architecture [45], using the Huggingface Transformers library [52] with the configurations shown in Table 5. BART is made out of a bidirectional encoder and an autoregressive decoder.

For the simplicity of the setup and because this is an entirely novel task for such a model (and thus, we wanted to establish a performance baseline), we decided to skip pre-training. After some preliminary training-performance tests, we settled on a context length of 2048 tokens, which allowed all our training to be done on an A100 GPU within a reasonable time (1 week). We note that this did limit the Lagrangians included in the dataset.

We chose cross-entropy loss for training our models because it is the default option and helps establish a performance baseline. Although cross-entropy loss does not inherently account for order-invariance in equations (as the order of terms in a Lagrangian or the order of symbols within a term is irrelevant if they are defined and contracted correctly), it turned out to be effective for our needs. Initially, we thought a custom loss function might be necessary to handle these nuances. However, as shown in this work, the performance with cross-entropy loss was already excellent, providing a strong benchmark for our novel task.

Table 5: BART configuration used in our work; in this setup, the only dropout is left at the default value, which applies to embeddings and feed-forward outputs rather than attention, activation, or classifier layers.

| Parameter | Value |
|---|---|
| Hidden size (Embedding Dimension) | 1024 |
| Number of encoder/decoder layers | 12 |
| Encoder/Decoder attention heads | 16 |
| Encoder/Decoder feed-forward dimension | 4096 |
| Max position embeddings | 2048 |
| Main dropout | 0.1 |
| Attention dropout | 0.0 |
| Activation dropout | 0.0 |
| Classifier dropout | 0.0 |
| Activation function | GELU |
| Initialization standard deviation | 0.02 |
| Scale embedding | False |
| Use cache | True |

## C  Tokenization examples

Table 7 to 11 illustrates example tokenizations of various term-types. The contraction information encodes how the field representations are contracted using the IDs. [''LORENTZ'', ''ID6'', ''ID4''] indicates that the indices related to the Lorentz group from symbols with ''ID6'' and ''ID4'' are contracted, namely $\overline{\sigma}_\mu$ and $D_\mu$. ['SU3'', ''ID9'', ''ID7''] indicates that the indices related to SU(3) group from the objects with ''ID9'' and ''ID7'' are contracted. As for the U(1) group, this is not necessary as invariance amounts to zero total charge in each term.

Table 6: Examples of interaction and mass terms in Lagrangians.

| Names | Terms |
|---|---|
| Trilinear term | $\lambda \phi_1 \phi_2 \phi_3$ |
| Quartic term | $\lambda \phi_1 \phi_2 \phi_3 \phi_4$ |
| Mass term (scalar) | $m^2 \phi^\dagger \phi$ |
| Mass term (fermion) | $m\, \psi_R^\dagger \chi_L + \text{h.c.}$ |
| Yukawa terms | $\lambda\, \psi_R^\dagger \phi \chi_L + \text{h.c.}$ |

Other than kinetic and mass terms, all other terms are generated by AutoEFT. Examples of possible interaction terms are shown in Table 6. As the output of AutoEFT is written using only fundamental indices of the internal symmetry groups [18], there will be instances where there are fields with more than one fundamental index. If the two indices of each of the two distinct fields are contracted, their IDs are repeated as shown in the SU(3) contractions

---

[18]We refer readers to Appendix A of AutoEFT for more details

897 in Table 10. Preliminary tests showed that removing these "double index" IDs proves to
898 be *distracting* for the transformer. As this introduces an extra task of understanding the
899 implicit deletion of an extra index, it is harder for the model to recognize the rules of
900 contractions.

Term : $i\,\psi_L^\dagger\,\overline{\sigma}^\mu D_\mu \psi_L$

| Information | Tokens |
|:---:|:---|
| $i$ | `i` |
| $\psi_L^\dagger(\mathbf{3},\,\mathbf{1},\,-3/4)$ | `FIELD, SPIN, 1, /, 2, SU3, 3, U1, -, 3, /, 4, HEL, 1, /, 2, DAGGER, ID9` |
| $\overline{\sigma}^\mu$ | `SIGMA_BAR, ID6` |
| $D_\mu^{SU(3)\times U(1)}$ | `DERIVATIVE, SU3, U1, ID4` |
| $\psi_L(\overline{\mathbf{3}},\,\mathbf{1},\,3/4)$ | `FIELD, SPIN, 1, /, 2, SU3, -, 3, U1, 3, /, 4, HEL, -, 1, /, 2, ID7` |
|  | `CONTRACTIONS` |
| $g^{\mu\nu}\psi_L^\dagger\,\overline{\sigma}_\mu\,D_\nu\,\psi_L$ | `LORENTZ, ID6, ID4` |
| $(\psi_L^\dagger)_{\dot\alpha}\,(\overline{\sigma}^\mu)^{\dot\alpha\beta}\,D_\mu\,(\psi_L)_\beta$ | `LORENTZ, ID6, ID9, ID7` |
| $(\psi_L^\dagger)^a\,\overline{\sigma}^\mu\,D_\mu\,(\psi_L)_a$ | `SU3, ID9, ID7` |

Table 7: Tokenization of an example kinetic term for a fermion field. The first column indicates the objects/information of the term that will be translated to tokens. The second column indicates the information in tokenized form. The example term here is thus encoded as a sequence of the tokens shown in the second column: `[i, FIELD, SPIN, 1,..., ID7, SU3, ID9, ID7 ]`

Term : $\partial^\mu\,\phi^\dagger\,\partial_\mu\phi$

| Information | Tokens |
|:---:|:---|
| $\partial^\mu$ | `DERIVATIVE, ID4` |
| $\phi^\dagger(\mathbf{1},\,\mathbf{1},\,0)$ | `FIELD, SPIN, 0, DAGGER, ID1` |
| $\partial_\mu$ | `DERIVATIVE, ID7` |
| $\phi(\mathbf{1},\,\mathbf{1},\,0)$ | `FIELD, SPIN, 0, ID5` |
|  | `CONTRACTIONS` |
| $g^{\mu\nu}\partial_\mu\,\phi^\dagger\,\partial_\nu\,\phi$ | `LORENTZ, ID4, ID7` |

Table 8: Tokenization of an example kinetic term for a singlet scalar field. The columns are as in Table 7

# D  Example cases

901

902 In this appendix, we present our evaluation of the sampled model's performance on
903 Lagrangians from existing particle physics models. The results are shown in Table 13. [19]

---

[19]Our training data does not include repeating fields, so all the Lagrangians discussed here involve only one generation.

$$\text{Term}: -\tfrac{1}{2}\text{Tr}\left([D_\mu, D_\nu][D^\mu, D^\nu]\right)_{\text{SU(2)}}$$

| Information | Tokens |
|---|---|
| — | - |
| $\tfrac{1}{2}\text{Tr}$ | |
| $[D_\mu, D_\nu]_{SU(2)}$ | COMMUTATOR_A, DERIVATIVE, SU2, ID6, COMMUTATOR_B, DERIVATIVE, SU2, ID5 |
| $[D^\mu, D^\nu]_{SU(2)}$ | COMMUTATOR_A, DERIVATIVE, SU2, ID1, COMMUTATOR_B, DERIVATIVE, SU2, ID0 |
| — | CONTRACTIONS |
| $g^{\mu\rho}g^{\nu\delta}[D_\mu, D_\nu][D_\rho, D_\delta]$ | LORENTZ, ID6, ID1 |
| $g^{\mu\rho}g^{\nu\delta}[D_\mu, D_\nu][D_\rho, D_\delta]$ | LORENTZ, ID5, ID0 |

Table 9: Tokenization of an example kinetic term for an SU(2) gauge fields. The columns are as in Table 7.

$$\text{Term}: \lambda\,\psi_L\,\chi_R^\dagger\,\phi$$

| Information | Tokens |
|---|---|
| $\lambda$ | |
| $\psi_L(\mathbf{1}, \mathbf{3}, 0)$ | FIELD, SPIN, 1, /, 2, SU2, 3, HEL, -, 1, /, 2, ID0 |
| $\chi_R^\dagger(\bar{\mathbf{3}}, \mathbf{3}, 0)$ | FIELD, SPIN, 1, /, 2, SU3, -, 3, SU2, 3, HEL, -, 1, /, 2, DAGGER, ID2 |
| $\phi(\mathbf{3}, \mathbf{3}, 0)$ | FIELD, SPIN, 0, SU3, 3, SU2, 3, ID6 |
| | CONTRACTIONS |
| $\epsilon^{\alpha_1\beta_1}\,(\psi_L)_{\alpha_1}\,(\chi_R^\dagger)_{\beta_1}\,\phi$ | LORENTZ, ID0, ID2 |
| $\epsilon^{b_1 b_2 c_1}\,(\psi_L)\,(\chi_R^\dagger)_{b_1 b_2}\,\phi_{c_1}$ | SU3, ID2, ID2, ID6 |
| $\epsilon^{a_1 b_2}\,(\psi_L)_{a_1 a_2}\,(\chi_R^\dagger)_{b_1 b_2}\,\phi_{c_1 c_2}$ | SU2, ID0, ID2 |
| $\epsilon^{a_2 c_1}\,(\psi_L)_{a_1 a_2}\,(\chi_R^\dagger)_{b_1 b_2}\,\phi_{c_1 c_2}$ | SU2, ID0, ID6 |
| $\epsilon^{b_1 c_2}\,(\psi_L)_{a_1 a_2}\,(\chi_R^\dagger)_{b_1 b_2}\,\phi_{c_1 c_2}$ | SU2, ID2, ID6 |

Table 10: Tokenization example of a Yukawa term. The columns are as in Table 7

904 The field content of these Lagrangians is detailed in Table 12. The invariance of Lagrangians
905 on the order of fields in input sequences was not explicitly enforced in our setup. This
906 means that in some cases the model's predictions can be different for different orderings of
907 the input fields. Therefore, we present both the best and mean performance scores derived
908 from all possible field orderings, as well as the main failure modes.

909     As observed in Section 5.1, the model's performance decreases with number of input
910 fields. The best predictions often have no errors, with most scoring at least 0.76. Mean scores
911 consistently exceed 0.72, with common failure modes primarily involving missing trilinear
912 Yukawa interactions (two fermions and one scalar). Although we included additional
913 trilinear terms in our dataset, most included only scalar fields. Yukawa terms require
914 an extra condition (conservation of spin), and even though we accounted for this in our
915 sampling, it was rare to encounter more than one or two Yukawa interactions in the dataset
916 Lagrangians. As a comparison, the set of fields in the Standard Model features the unique
917 property of having six Yukawa terms. As a result, the model struggles to generate the

Term : $\lambda\,\phi\,\phi^\dagger\,\Phi\,\Phi^\dagger$

| Information | Tokens |
|---|---|
| $\lambda$ | |
| $\phi(\mathbf{3},\,\mathbf{2},\,6)$ | FIELD, SPIN, 0, SU3, 3, SU2, 3, U1, 6, ID5 |
| $\phi^\dagger(\overline{\mathbf{3}},\,\mathbf{2},\,-6)$ | FIELD, SPIN, 0, SU3, -, 3, SU2, 3, U1, -, 6, DAGGER, ID7 |
| $\Phi(\overline{\mathbf{3}},\,\mathbf{1},\,-7/2)$ | FIELD, SPIN, 0, SU3, -, 3, U1, -, 7, /, 2, ID8 |
| $\Phi^\dagger(\mathbf{3},\,\mathbf{1},\,7/2)$ | FIELD, SPIN, 0, SU3, 3, U1, 7, /, 2, DAGGER, ID2 |
| | CONTRACTIONS |
| $\epsilon^{a_1 b_2 c_2}\,\phi_{a_1}\,\phi^\dagger{}_{b_1 b_2}\,\Phi_{c_1 c_2}\,\Phi^\dagger{}_{d_1}$ | SU3, ID5, ID7, ID8 |
| $\epsilon^{b_1 c_1 d_1}\,\phi_{a_1}\,\phi^\dagger{}_{b_1 b_2}\,\Phi_{c_1 c_2}\,\Phi^\dagger{}_{d_1}$ | SU3, ID7, ID8, ID2 |
| $\epsilon^{a_1 b_1}\,\phi_{a_1 a_2}\,\phi^\dagger{}_{b_1 b_2}\,\Phi\,\Phi^\dagger$ | SU2, ID5, ID7 |
| $\epsilon^{a_2 b_2}\,\phi_{a_1 a_2}\,\phi^\dagger{}_{b_1 b_2}\,\Phi\,\Phi^\dagger$ | SU2, ID5, ID7 |

Table 11: Tokenization of an example scalar quadratic term. The columns are as in Table 7.

one-generation Standard Model as it tends to miss four out of the six Yukawa terms. A tailored enrichment of a new training dataset with specific combinations of fields that lead to many Yukawa terms, or simply a fine-tuning of our trained model with some examples would likely resolve this issue. We leave that for future work.

Notably, in some scalar extensions (2-Higgs Doublet Model (2HDM) and Georgi-Machacek Model), the model often predicts extra terms. Specifically, in the case of the 2HDM, the model generated the correct Lagrangian although with repeated terms.

Table 12: Field content of example models used.

| Model name | Field content |
|---|---|
| Scalar QED | $\phi(\mathbf{1},\mathbf{1},1)$ |
| Scalar Weak | $\phi(\mathbf{1},\mathbf{2},0)$ |
| Scalar QCD | $\phi(\mathbf{3},\mathbf{1},0)$ |
| Scalar EW | $\phi(\mathbf{1},\mathbf{2},1)$ |
| SM-Gauged Scalar | $\phi(\mathbf{3},\mathbf{2},1)$ |
| Lepton Sector | $L_L(\mathbf{1},\mathbf{2},-1)\ ,\ e_R(\mathbf{1},\mathbf{1},-2)$ |
| Quark Sector | $Q_L(\mathbf{3},\mathbf{2},\frac{1}{3})\ ,\ u_R(\mathbf{3},\mathbf{1},\frac{4}{3})\ ,\ d_R(\mathbf{3},\mathbf{1},-\frac{2}{3})$ |
| Higgs and Lepton Sector | $L_L(\mathbf{1},\mathbf{2},-1)\ ,\ e_R(\mathbf{1},\mathbf{1},-2)\ ,\ H(\mathbf{1},\mathbf{2},1)$ |
| Higgs and Quark Sector | $Q_L(\mathbf{3},\mathbf{2},\frac{1}{3})\ ,\ u_R(\mathbf{3},\mathbf{1},\frac{4}{3})\ ,\ d_R(\mathbf{3},\mathbf{1},-\frac{2}{3})\ ,\ H(\mathbf{1},\mathbf{2},1)$ |
| Standard Model (SM) | $L_L(\mathbf{1},\mathbf{2},-1)\ ,\ e_R(\mathbf{1},\mathbf{1},-2)\ ,\ Q_L(\mathbf{3},\mathbf{2},\frac{1}{3})\ ,\ u_R(\mathbf{3},\mathbf{1},\frac{4}{3})\ ,\ d_R(\mathbf{3},\mathbf{1},-\frac{2}{3})\ ,\ H(\mathbf{1},\mathbf{2},1)$ |
| Scalar Leptoquark, $\Phi_1$ | $\Phi_1(\mathbf{3},\mathbf{1},-\frac{8}{3})\ ,\ d_R(\mathbf{3},\mathbf{1},-\frac{2}{3})\ ,\ e_R(\mathbf{1},\mathbf{1},-2)$ |
| Scalar Leptoquark, $\Phi_2$ | $\Phi_2(\mathbf{3},\mathbf{2},\frac{1}{3})\ ,\ d_R(\mathbf{3},\mathbf{1},-\frac{2}{3})\ ,\ L_L(\mathbf{1},\mathbf{2},-1)$ |
| Scalar Leptoquark, $\Phi_3$ | $\Phi_3(\mathbf{3},\mathbf{3},\frac{2}{3})\ ,\ Q_L(\mathbf{3},\mathbf{2},\frac{1}{3})\ ,\ L_L(\mathbf{1},\mathbf{2},-1)$ |
| Scalar Leptoquark, $\Phi_1'$ | $\Phi_1'(\mathbf{3},\mathbf{1},-\frac{2}{3})\ ,\ Q_L(\mathbf{3},\mathbf{2},\frac{1}{3})\ ,\ L_L(\mathbf{1},\mathbf{2},-1)\ ,\ u_R(\mathbf{3},\mathbf{1},\frac{4}{3})\ ,\ e_R(\mathbf{1},\mathbf{1},-2)$ |
| Scalar Leptoquark, $\Phi_2'$ | $\Phi_2'(\mathbf{3},\mathbf{2},\frac{7}{3})\ ,\ Q_L(\mathbf{3},\mathbf{2},\frac{1}{3})\ ,\ L_L(\mathbf{1},\mathbf{2},-1)\ ,\ u_R(\mathbf{3},\mathbf{1},\frac{4}{3})\ ,\ e_R(\mathbf{1},\mathbf{1},-2)$ |
| 2HDM | $H_1(\mathbf{1},\mathbf{2},1)\ ,\ H_2(\mathbf{1},\mathbf{2},-1)$ |
| Georgi-Machacek Model [53] | $H(\mathbf{1},\mathbf{2},1)\ ,\ \Delta(\mathbf{1},\mathbf{3},0)\ ,\ \Delta_C(\mathbf{1},\mathbf{3},2)$ |
| RPV-MSSM LLE-Sector | $\tilde{L}_L(\mathbf{1},2,-1)\ ,\ \tilde{e}_R(\mathbf{1},\mathbf{1},-2)\ ,\ L_L(\mathbf{1},\mathbf{2},-1)\ ,\ e_R(\mathbf{1},\mathbf{1},-2)$ |
| RPV-MSSM UDD-Sector | $\tilde{u}_R(\mathbf{3},\mathbf{1},\frac{4}{3})\ ,\ \tilde{d}_R(\mathbf{3},\mathbf{1},-\frac{2}{3})\ ,\ u_R(\mathbf{3},\mathbf{1},\frac{4}{3})\ ,\ d_R(\mathbf{3},\mathbf{1},-\frac{2}{3})$ |
| RPV-MSSM LQD-Sector | $\tilde{L}_L(\mathbf{1},\mathbf{2},-1)\ ,\ \tilde{q}_L(\mathbf{3},\mathbf{2},\frac{1}{3})\ ,\ \tilde{d}_R(\mathbf{3},\mathbf{1},-\frac{2}{3})\ ,\ L_L(\mathbf{1},\mathbf{2},-1)\ ,\ Q_L(\mathbf{3},\mathbf{2},\frac{1}{3})\ ,\ d_R(\mathbf{3},\mathbf{1},-\frac{2}{3})$ |

Table 13: Performance on various models. Score of Best prediction, Mean score and its corresponding failure modes are shown here.

| Model name | $n_{\text{Fields}}$ | Best prediction | | | Overall prediction (mean) | | | Main failure mode |
|---|---|---|---|---|---|---|---|---|
| | | $S_{\text{Lagrangian}}$ | $S_{\text{Correct}}$ | $P_{\text{Length}}$ | $S_{\text{Lagrangian}}$ | $S_{\text{Correct}}$ | $P_{\text{Length}}$ | |
| Scalar QED | 1 | 1 | 1 | 0 | 1 | 1 | 0 | - |
| Scalar weak | 1 | 1 | 1 | 0 | 1 | 1 | 0 | - |
| Scalar QCD | 1 | 1 | 1 | 0 | 1 | 1 | 0 | - |
| Scalar EW | 1 | 1 | 1 | 0 | 1 | 1 | 0 | - |
| SM-gauged scalar | 1 | 1 | 1 | 0 | 1 | 1 | 0 | - |
| Lepton sector | 2 | 1 | 1 | 0 | 1 | 1 | 0 | - |
| Quark sector | 3 | 1 | 1 | 0 | 1 | 1 | 0 | - |
| Higgs + lepton sector | 3 | 1 | 1 | 0 | 0.98 | 0.98 | 0 | Incorrect charge assignments |
| Higgs + quark sector | 4 | 0.86 | 0.86 | 0 | 0.8 | 0.8 | 0 | Missing Yukawas |
| Standard Model | 6 | 0.77 | 0.77 | 0 | 0.74 | 0.74 | 0 | Missing Yukawas |
| Scalar LQ, $\Phi_1$ | 3 | 1 | 1 | 0 | 0.86 | 0.86 | 0 | Missing Yukawas |
| Scalar LQ, $\Phi_2$ | 3 | 1 | 1 | 0 | 1 | 1 | 0 | - |
| Scalar LQ, $\Phi_3$ | 3 | 1 | 1 | 0 | 0.87 | 1 | 0.13 | Extra Yukawas |
| Scalar LQ, $\Phi_1'$ | 5 | 0.76 | 0.76 | 0 | 0.76 | 0.76 | 0 | Missing Yukawas |
| Scalar LQ, $\Phi_2'$ | 5 | 0.88 | 0.88 | 0 | 0.79 | 0.79 | 0 | Missing Yukawas |
| 2HDM | 2 | 0.81 | 1 | 0.19 | 0.77 | 0.96 | 0.19 | Extra Quartic and mass terms |
| Georgi-Machacek | 3 | 0.90 | 0.93 | 0.03 | 0.76 | 0.76 | 0.01 | Extra Quartic and Missing Quartic and trilinears |
| RPV-MSSM LLE | 4 | 1 | 1 | 0 | 0.90 | 0.90 | 0 | Incorrect field / Incorrect dagger |
| RPV-MSSM UDD | 4 | 0.88 | 0.88 | 0 | 0.84 | 0.84 | 0 | Missing Yukawas |
| RPV-MSSM LQD | 6 | 0.78 | 0.78 | 0 | 0.72 | 0.72 | 0 | Missing Yukawas and scalar trilinears |

# E  OOD performance using non-minimal fractions for U(1) charges

In this appendix, we evaluate the sampled model's performance on Lagrangians with OOD U(1) charges. In our training dataset, as discussed in 3.2, all fractional U(1) charges are in their simplest form. This would mean that all non-minimal fractions for representing charges as shown in Table 14 can be used as OOD scenarios.[20].

To test this, we generated a new dataset with fields that have such OOD charges by modifying Lagrangians from the sampled datasets (both training and test/evaluation datasets). For 1-field scenarios, we go through all possible fields that can have OOD charges, which amounts to 1404 OOD Lagrangians. For n-field scenarios where only one field has OOD charge, we generated 10 cases for each OOD charge with all other quantum numbers chosen at random. We included 1040 such Lagrangians per n-field scenario. For each n-field scenario with more than one OOD Field, we generated 1000 Lagrangians by sampling Lagrangians from the training and test dataset. As it is more difficult to find Lagrangians where many of the fields can be modified to have OOD charges, for larger n, we sometimes take copies of the same Lagrangian with different OOD representations of the charges.

Table 15 shows the percentage of reasonable predicted OOD Lagrangians[21]. Similar to Section 5.1, almost all cases are reasonable (>97%). Table 17 and Table 16 show the percentage of predicted OOD Lagrangians with correct charge assignments and the percentage of predicted OOD Lagrangians with fully conserved U(1), respectively. We do this as the model can write the wrong charges in terms that, in fact, conserve U(1)

---

[20]As the model has never seen 0 in fractions nor one (1) as integer, they will not be considered as part of the OOD charges

[21]We explain what reasonable means in Section 5.1

symmetry. We can see that the predictions worsen with more OOD fields and more input fields. However, it is worth pointing out that in scenarios with more than 3 OOD Fields, there tends to be more U(1) conserving Lagrangians than Lagrangians with correct charge assignments. This indicates that while the model fails to grasp many OOD charges, it will still try to enforce U(1) conservation.

Table 18 shows the accuracy of the model on OOD Lagrangians (the percentage of Lagrangians with a Lagrangian score of 1). As expected, the model performs worse with more OOD fields and input fields. This can be attributed to the model's poor ability to understand numbers. This is illustrated in Figure 11 where Lagrangians with and without trilinears are considered separately. With InD accuracy as a benchmark, there is a severe decrease in OOD accuracy when we only focus on Lagrangians with trilinears. The model struggles at predicting Lagrangians that contain trilinears and all OOD fields, showing then a 20% accuracy. On the other hand, the model still performs reasonably well on OOD Lagrangians without trilinears, with some cases slightly outperforming InD scenarios. This is not surprising since for all mass terms and most quartic terms, the model learned to pair up fields that have the same charges without any understanding of arithmetic (as $a - a + b - b = 0$). When an integer U(1) charge get modified to an OOD fractional charge, the model also performs better, as fractional charges are better represented in the training data. Meanwhile, all trilinear terms and some quartic terms require an understanding of arithmetic (e.g. $a + b + c = 0$) to write all possible U(1) conserving terms.

Text-based encoding schemes for numbers, how we encoded U(1) charges in this work, are prone to take advantage of spurious correlations in the data and perform worse in OOD dataset, as shown in Reference [54]. In OOD Lagrangians with trilinears, it is likely that the model memorized possible charge combinations that would lead to trilinear terms instead of learning how to sum up charges to conserve U(1). A continuous number encoding approach would have helped in this scenario, and will be the subject of future work.

Table 14: U(1) charges conversion table. The model has not seen cases with zero as the numerator nor one as the denominator and those cases are not considered here.

| InD charges | OOD charges |
| --- | --- |
| 1 | 2 / 2, 3 / 3, 4 / 4, 5 / 5, 6 / 6, 7 / 7, 8 / 8, 9 / 9 |
| 1 / 2 | 2 / 4, 3 / 6, 4 / 8 |
| 1 / 3 | 2 / 6, 3 / 9 |
| 1 / 4 | 2 / 8 |
| 2 | 4 / 2, 6 / 3, 8 / 4 |
| 2 / 3 | 4 / 6, 6 / 9 |
| 3 | 6 / 2, 9 / 3 |
| 3 / 2 | 6 / 4, 9 / 6 |
| 3 / 4 | 6 / 8 |
| 4 | 8 / 2 |
| 4 / 3 | 8 / 6 |

# F   Vector offset method

In natural language processing (NLP), the *vector offset method* is often used to perform analogy tasks [51]. The method assumes that relationships between words can be expressed

Table 15: Percentage of reasonable OOD Lagrangians

| Number of fields | Number of OOD fields | | | | | |
|---|---|---|---|---|---|---|
| | 1 | 2 | 3 | 4 | 5 | 6 |
| 1 | 100 % | | | | | |
| 2 | 100 % | 100 % | | | | |
| 3 | 99.9 % | 99.9 % | 99.6 % | | | |
| 4 | 99.7 % | 98.9 % | 99.2 % | 98.9 % | | |
| 5 | 99.9 % | 99.5 % | 99.2 % | 98.9 % | 99.2 % | |
| 6 | 99 % | 99.2 % | 98.8 % | 98.9 % | 96.7 % | 97.7 % |

Table 16: Percentage of OOD Lagrangians with correct charge assignment

| Number of fields | Number of OOD fields | | | | | |
|---|---|---|---|---|---|---|
| | 1 | 2 | 3 | 4 | 5 | 6 |
| 1 | 100 % | | | | | |
| 2 | 99.8 % | 97.6 % | | | | |
| 3 | 98.9 % | 98 % | 94.6 % | | | |
| 4 | 98.1 % | 97.4 % | 93.8 % | 91.5 % | | |
| 5 | 97.1 % | 95.1 % | 90.9 % | 85.4 % | 82.4 % | |
| 6 | 93.8 % | 90.6 % | 84.1 % | 77.7 % | 73.7 % | 65.3 % |

Table 17: Percentage of U(1) conserving OOD Lagrangians

| Number of fields | Number of OOD fields | | | | | |
|---|---|---|---|---|---|---|
| | 1 | 2 | 3 | 4 | 5 | 6 |
| 1 | 100 % | | | | | |
| 2 | 99.9 % | 99.5 % | | | | |
| 3 | 98.7 % | 98.5 % | 94.5 % | | | |
| 4 | 95.2 % | 95.1 % | 93 % | 91 % | | |
| 5 | 93.9 % | 96 % | 93 % | 89.7 % | 87.7 % | |
| 6 | 92 % | 95.2 % | 92.8 % | 88.1 % | 86.1 % | 79.7% |

through vector arithmetic in the embedding space, where regularities can be observed as constant vector offsets between pairs of words sharing a particular relationship.

This concept is illustrated with the well-known example: "*king is to man as queen is to woman*", which can be mathematically expressed as:

$$\vec{v}_{\text{king}} - \vec{v}_{\text{man}} \approx \vec{v}_{\text{queen}} - \vec{v}_{\text{woman}} \tag{F.1}$$

where $\vec{v}_w$ is the embedding vector of word $w$.

By defining:

Table 18: Percentage of Lagrangians with perfect prediction (Accuracy)

| Number of fields | Number of OOD fields | | | | | |
|---|---|---|---|---|---|---|
| | 1 | 2 | 3 | 4 | 5 | 6 |
| 1 | 100 % | | | | | |
| 2 | 93.1 % | 53.1 % | | | | |
| 3 | 84.8 % | 80.9 % | 54 % | | | |
| 4 | 80.4 % | 83.9 % | 66.1 % | 48.5 % | | |
| 5 | 77.4 % | 79.9 % | 66.6 % | 52.2 % | 40.7 % | |
| 6 | 70.5 % | 74.6 % | 61.5 % | 49.2 % | 39.9 % | 28.8 % |

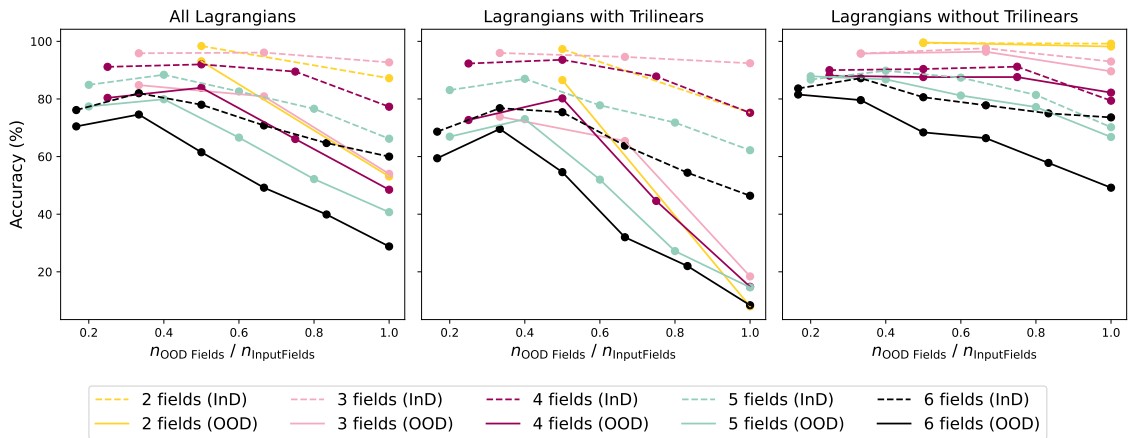

Figure 11: Accuracy (Percentage of Lagrangians with perfect prediction) with varying OOD Fields to Input Fields ratios. **Left**: Accuracy on full OOD dataset.**Middle**: Accuracy on OOD datasets with Lagrangians containing trilinear terms. **Right**: Accuracy on OOD dataset with Lagrangians without trilinear terms.

$$\vec{v}_1^{\text{Royal}} = \vec{v}_{\text{king}} - \vec{v}_{\text{man}} \tag{F.2}$$

$$\vec{v}_2^{\text{Royal}} = \vec{v}_{\text{queen}} - \vec{v}_{\text{woman}} \tag{F.3}$$

If the relationship is consistent, we expect:

$$\vec{v}_1^{\text{Royal}} \approx \vec{v}_2^{\text{Royal}} \tag{F.4}$$

However, due to the complexity of language and high dimensionality of embeddings, these are often approximate. Cosine similarity is then used to evaluate the closeness between vectors:

$$s(\vec{v}_i, \vec{v}_j) = \frac{\vec{v}_i \cdot \vec{v}_j}{\|\vec{v}_i\|\|\vec{v}_j\|} \tag{F.5}$$

where $s = +1$ for two proportional vectors, $s = 0$ for orthogonal vectors, and $s = -1$ for opposite vectors.

## F.1    Application to field conjugation

Applying this method to our context, we define the conjugation vector for each field as:

$$\vec{v}_{C_i} = \vec{v}_{F_i} - \vec{v}_{F_i^C} \tag{F.6}$$

where:

- $F_i$ is an arbitrary field with quantum numbers $\{Q_i\}$,

- $F_i^C$ is the conjugate of $F_i$,

- and $\vec{v}_f$ is the embedding vector of a field $f$.

We would then expect that:

$$\vec{v}_{F_i} - \vec{v}_{F_i^C} \approx \vec{v}_{F_j} - \vec{v}_{F_j^C} \tag{F.7}$$

i.e., the conjugation vectors $\vec{v}_{C_i}$ should be similar for different fields, indicating a consistent representation of the conjugation operation in the embedding space.

To assess this, we calculate the absolute value of the cosine similarity between all pairs of conjugation vectors $\vec{v}_{C_i}$ and $\vec{v}_{C_j}$:

$$|s(\vec{v}_{C_i}, \vec{v}_{C_j})| \tag{F.8}$$

As a reference, we also compute difference vectors for randomly paired fields:

$$\vec{v}_{R_i} = \vec{v}_{F_i} - \vec{v}_{F_i^R} \tag{F.9}$$

where $\vec{v}_{F_i^R}$ is randomly chosen from all possible fields with no predefined relation to $\vec{v}_{F_i}$. We then compute the absolute cosine similarities between these random translation vectors.

By comparing the distributions of $|s(\vec{v}_{C_i}, \vec{v}_{C_j})|$ and $|s(\vec{v}_{R_i}, \vec{v}_{R_j})|$, we can determine whether the model has learned the conjugation operation as a consistent transformation in the embedding space. A peak near 1 in the distribution of conjugation vector similarities would indicate a preferred axis, while a flat distribution for random translations would confirm that this is not a general feature of the embedding space. The results of this analysis are presented in Figure 10 in the main text.

## F.2    Details on Conjugation Transformations

To understand why the conjugation transformation appears more defined for fermions than for scalars, we analyzed the transformation vectors associated with individual symmetry properties within the fields. We computed conjugation vectors specific to U(1) charge $\vec{v}_{C_i}^{\mathrm{U}(1)}$, SU(3) representation $\vec{v}_{C_i}^{\mathrm{SU}(3)}$, and helicity $\vec{v}_{C_i}^{H}$, by conjugating only the respective group charges.

In Figure 12 (Left), analysis of the absolute cosine similarities $|s|$ between pairs of these vectors reveals distinct behaviors. Both SU(3) conjugation vectors and helicity conjugation vectors exhibit strong internal consistency, with their respective similarity distributions ($|s(\vec{v}_{C_i}^{\mathrm{SU}(3)}, \vec{v}_{C_j}^{\mathrm{SU}(3)})|$ and $|s(\vec{v}_{C_i}^{H}, \vec{v}_{C_j}^{H})|$) peaking sharply near $|s| = 1$. This indicates well-defined transformation axes for these properties. In contrast, the similarity distribution for $U(1)$ conjugation vectors $|s(\vec{v}_{C_i}^{U(1)}, \vec{v}_{C_j}^{U(1)})|$ is broad, indicating a less structured transformation axis.

This difference likely stems from the dataset composition, where the main variations among the unique fields relate to their U(1) charges. This might make it challenging for the

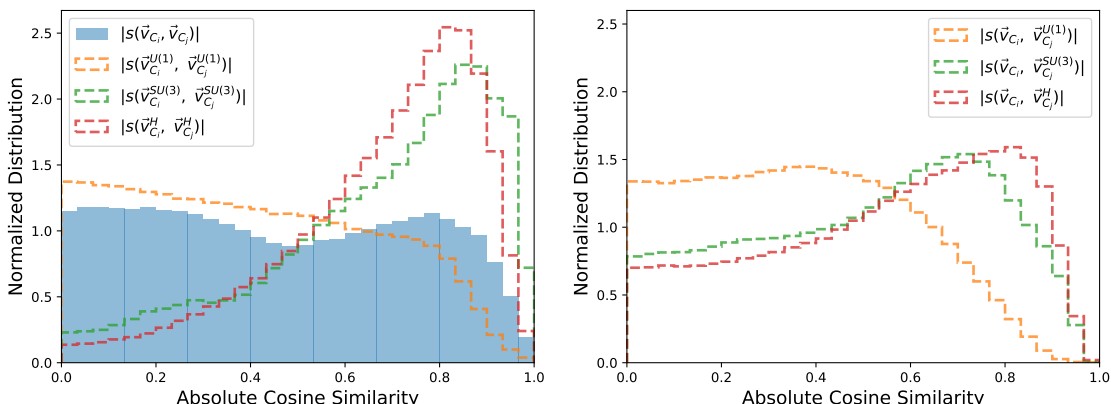

Figure 12: Normalized distribution of absolute cosine similarities between: (**Left**) pairs of conjugation vectors of different symmetry group. (**Right**) conjugation vectors of different symmetry group and the overall conjugation vectors.

model to isolate a specific U(1) conjugation axis distinct from general variations between fields.

Figure 12 (Right) further illustrates the relationship between these symmetry-specific axes and the overall conjugation axis. Both the SU(3) and helicity axes align closely with the overall conjugation axis, whereas the U(1) axis appears misaligned and more weakly defined (eventhough it is not perpendicular). With this in mind, this suggests that the well-defined conjugation axis observed for fermions is driven primarily by two well-defined and co-aligned axes (SU(3) and helicity) with the less coherent U(1) component contributes additional variability. Scalars, which lack a helicity axis, thus exhibit a less pronounced overall conjugation direction.

Given that conjugation of fields often involves changes of only a few tokens (e.g., addition/removal of a "‑" sign for U(1) and SU(3) conjugation, helicity change, etc.), we also examined the distributions when the number of token differences between two random fields is low. We defined several control vector types representing such minimal changes:

$\vec{v}_{R_i}^{-}$    : Random fields where the "‑" token is part of the token difference.

$\vec{v}_{R_i}^{\Delta}$    : Random fields where there are less than 3 token differences.

$\vec{v}_{R_i}^{-,\Delta}$    : Random fields where there are less than 3 token differences and the "‑" token are one of them.

$\vec{v}_{M_i}$    : Random insertion or removal of a "‑" token into a field.

The distributions of the absolute cosine similarities between pairs of these minimal-change vectors (Figure 13) showed that most resulting distributions peak sharply at $|s| = 0$. The exception involved comparisons between random field differences that happen to include the "‑" token ($\vec{v}_{R_i}^{-}$), which displayed a broader structure somewhat similar to general random field differences. This confirms that the distinct conjugation axes identified previously and in Section 6 represent learned transformations linked to physical properties, rather than superficial effects stemming solely from minimal token edits.

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
