# Peer review of "Generating particle physics Lagrangians with transformers"

_SciPost Physics_

## Round 1 · Referee Report · Anonymous (Referee 1) · 2025-8-24

Strengths

I like the clear explanation of how LLM-concepts are applied to a particle physics theory problem.

In particular the out of domain-section is clearly written and explains the method well. Quantification of abstract similarity is particularly nicely carried out.

Weaknesses

1 - While interesting in its own right, I am wondering how, on the longer term, the Lagrangian-generating AI might be applied: To my understanding it generates EFT-Lagrangians, and would it be imaginable that subsequent processing determines for instance S-matrix elements and scattering amplitudes?

2 - Many intellectual achievements in the construction of the standard model of particle physics seem to be implicitly assumed in the proposed generative AI, or are beyond the scope of what the algorithm would be able to provide: Could you comment on this, in particular with these two points in mind?

2a - Many choices in the construction of Lagrangians are implicit: restriction to squares of first derivatives in the kinetic terms disallow derivative couplings or Ostrogradsky-unstable theories, explicitly constructed differences between left- and right-handed fermion doubles already codify P-violation. Can you please be more explicit in what implicit or hidden assumptions are at the basis of the Lagrange-construction?

2b - Similarly, adding more particles for instance with SU(3) symmetries would have them co-exist with possible interactions - but there would not be a larger symmetry group accommodating them, right?

3 - There are studies for automatic generation of Lagrange-functions from data through symbolic regression. Can you please add a comment on how the two are related? To my understanding, your methods looks for mathematical consistency, and tying in to my first remark, does not take direct input from experimental data?

4 - Similarly, it'd be good to emphasise that the gauge particle content and their group structure in this work is fixed, and the matter particle content is variable. On a related note - there is no concept of quark mixing, right?

5 - Would the generative AI find Lagrange-densities with V-A coupling, i.e. explicit parity violation?

Report

The paper demonstrates parallels between the construction of viable Lagrange densities for application in particle physics by generative AI and language models. I like how concepts like tokenization are applied to the construction of mathematical objects. I find the investigations presented interesting, as they demonstrate well what the authors had in mind.

The decision to relegate the technical parts to the appendices is understandable, but did not help me to access the paper. It was only after working through the appendix that things became apparent. I'd argue that the appendices are vital to the understanding of the paper.

Requested changes

Please take account of the points listed under "weakness" in the evaluation. Further points would be:

  1. I find it difficult to assess whether the number of Lagrange densities for instance in the training data set is large or not. Can you make a combinatorial argument what fraction of Lagrange densities is contained?

  2. I've been playing with the online tool for Lagrangian generation, which I find a great addition to the paper and a prime example of accessibility. Would it be imaginable that the coupling terms are non-polynomial?

  3. Could you please explain how you quantify "hallucination" and what measures to take against it?

  4. Would the boundary between well-constructed Lagrange-densities and faulty ones as a function of the number of fields (discussed in the OOD-section) shift towards more fields if more fields are contained in the training data? In other words, is there anything peculiar about 6 fields?

Recommendation

Ask for minor revision

---

## Editorial Decision

in_refereeing